

# Landscape response to tectonic deformation and cyclic climate change since ca. 800 ka in the southern Central Andes

Elizabeth N. Orr[1,2*], Taylor F. Schildgen[1,3], Stefanie Tofelde[4], Hella Wittmann[1] and Ricardo N. Alonso[5]

1: GFZ German Research Centre for Geosciences, Telegrafenberg, 14473 Potsdam, Germany
2: Department of Geography, Durham University, Durham, DH1 3LE, United Kingdom
3: Institute for Geosciences, University of Potsdam, Karl-Liebknecht-Str. 24-25, 14476 Potsdam, Germany
4: Institute of Geological Sciences, Freie Universität Berlin, 12249 Berlin, Germany
5: Facultad de Ciencias Naturales, Universidad Nacional de Salta, Salta, 4400 Argentina

*Corresponding author: Elizabeth N. Orr (elizabeth.orr2@durham.ac.uk)

**Abstract**

Theory suggests that the response time of alluvial channel long-profiles to perturbations in climate is related to the magnitude of the forcing and the length of the system. Shorter systems may record a higher frequency of forcing compared to longer systems. Empirical field evidence that system length plays a role in the climate periodicity preserved within the sedimentary record is, however, sparse. The Toro Basin in the Eastern Cordillera of NW Argentina provides an opportunity to test these theoretical relationships as this single source-to-sink system contains a range of sediment deposits, located at varying distances from the source. A suite of eight alluvial fan deposits is preserved along the western flanks of the Sierra de Pascha. Farther downstream, a flight of cut-and-fill terraces have been linked to eccentricity-driven (100-kyr) climate cycles since ca. 500 ka. We applied cosmogenic radionuclide ($^{10}$Be) exposure dating to the fan surfaces to explore (1) how channel responses to external perturbations may or may not propagate downstream, and (2) the differences in landscape response to forcing frequency as a function of channel length. We identified two generations of fan surfaces: the first (G1) records surface activity and abandonment between ca. 800 and 500 ka and the second (G2) within the last 100 kyr. G1 fans record a prolonged phase of net incision, which has been recognised throughout the Central Andes, and was likely triggered by enhanced 100-kyr global glacial cycles following the Mid-Pleistocene Transition. Relative fan surface stability followed, while 100-kyr cut-and-fill cycles occurred downstream, suggesting a disconnect in behaviour between the two channel reaches. G2 fans record higher frequency climate forcing, possibly the result of precessional forcing of climate (ca. 21/40-kyr timescales). The lack of a high-frequency signal farther downstream provides field support for theoretical predictions of a filtering of high-frequency climate forcing with increasing channel length. We show that multiple climate periodicities can be preserved within the sedimentary record of a single basin. Differences in the timing of alluvial fan and fluvial terrace development in the Toro Basin appears to be associated with how channel length affects fluvial response times to climate forcing as well as local controls on net incision, such as tectonic deformation.



**Plain Language Summary**

Fluvial terraces and alluvial fans in the Toro Basin, NW Argentina record river evolution and global climate cycles over time. Landform dating reveals lower-frequency climate cycles (100-kyr) preserved downstream and higher-frequency cycles (21/40-kyr) upstream, supporting theoretical predications that longer rivers filter out higher-frequency climate signals. This finding improves our understanding of the spatial distribution of sedimentary paleoclimate records within landscapes.

## 1. Introduction

Fluvial landforms, sediment deposits and the channel form of alluvial systems can be used to reveal landscape response to past environmental change (Castelltort and Van Den Driessche, 2003, Godard et al., 2013; Dey et al., 2016; Romans et al., 2016; Mescolotti et al., 2021). Alluvial channels respond to climate or tectonic driven changes in water discharge, sediment discharge, or base level elevation by adjusting at least one of their characteristics: bed slope, channel width, channel depth, sediment transport rates or grain-size distribution (Mackin 1948; Savi et al., 2020). We can observe this channel adjustment via sediment aggradation or incision events, which modify channel bed elevations (Howard, 1982; van den Berg et al., 2008; Wickert and Schildgen, 2019; Tofelde et al., 2019). Fluvial landforms such as terraces and alluvial fans, which develop along these channels because of this aggradation or incision, can provide a useful record of how the alluvial-channel system has evolved over time (Rohais et al., 2012; Armitage et al., 2013; Kober et al., 2013; Counts et al., 2015; Mather et al., 2017; Tofelde et al., 2021).

Theory suggests that the time required for an alluvial-channel long profile to adjust to a change in climate forcing (response time) varies with the magnitude and type of the forcing (sediment supply versus water supply) and the length of the system; shorter systems respond faster and, hence, may record a higher frequency of forcing compared to longer systems (Paola et al., 1992; Castelltort and Van Den Driessche, 2003; Godard et al., 2013; McNab et al., 2023). The length scale over which periodic forcing delivered at the channel head affects the channel long profile is proportional to the square root of the period of the forcing (Paola et al., 1992), which means that higher frequency forcing is filtered out with distance downstream. Evidence of this relationship is preserved in several sedimentary basins in the Central Andes. Tributary catchments of the Humahuaca Basin (23°S) retain late Quaternary fluvial deposits between 10 and 100 km downstream from the basin headwaters, which record precessional (21 kyr) cycles in aggradation and incision (Schildgen et al., 2016). In the Toro Basin (24.5°S), a flight of fluvial cut-and-fill terraces with periodicity of 100-kyr has been linked to eccentricity-driven climate change (Tofelde et al., 2017). These terraces have an upstream channel length of ~60–80 km. Pliocene-



Late Pleistocene sediment deposits are preserved ~140–160 km downstream from the headwaters of
the Iruya Basin (22°S) of the northern Central Andes and record long eccentricity (400-kyr) cycles
(Fisher et al., 2023). Crucially, only a single climate periodicity has been recorded in each these basins
to date. To further test this theoretical relationship between climate periodicity and system length, we
aim to investigate whether multiple periodicities can be preserved within a single basin, and if this is
the case, whether higher frequency climate forcing is only observed in the uppermost reaches of the
basin.

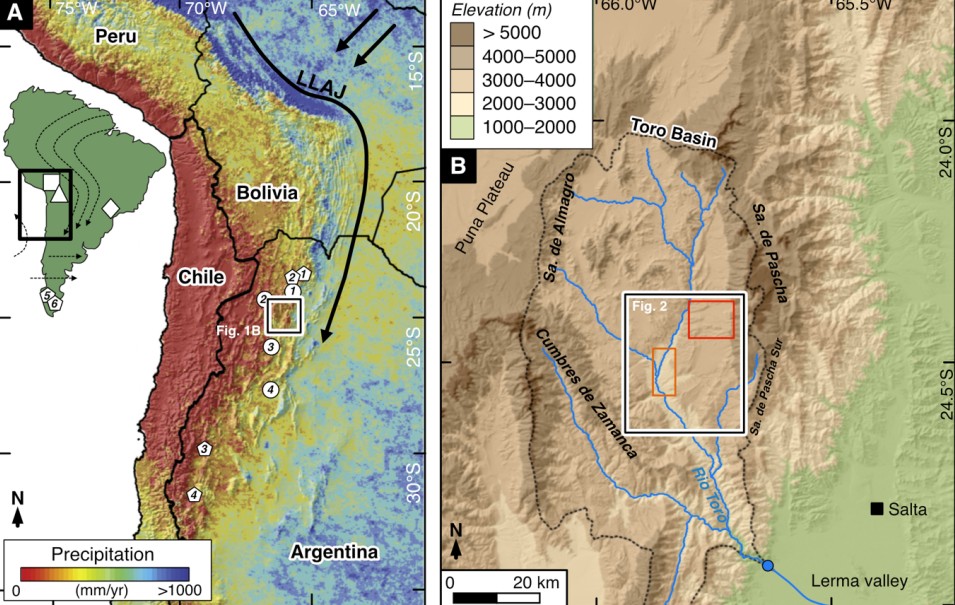

**Figure 1.** Overview of the topography, rainfall and moisture transport of the Central Andes. A) TRMM2B31 rainfall map
(Bookhagen and Strecker, 2008). Moisture is transported (black arrows) from Atlantic sources during the SASM by the Low-
Level Andean Jet (LLAJ; Vera et al., 2006). The Toro Basin is outlined by the white-black bordered box. Circle symbols
denote regional glacial record locations: (1) Nevado de Chañi (24.0°S, 65.7°W; Martini et al., 2017), (2) Quevar Volcano
(24.4°S, 66.8°W; Luna et al., 2018), (3) Sierra de Quilmes (26.2°S, 66.2°W; Zech et al., 2017) and the (4) Sierra de Aconquija
(27.2°S, 66.1°W; D'Arcy et al., 2019a). Pentagon symbols denote MPT geomorphic record locations: (1) Casa Grande Basin
(23°S, 66.5°W; Pingel et al., 2019b), (2) Salinas Grandes Basin (23.5°S, 66°W; Pingel et al., 2019b), (3) Iglesia Basin (30.5°S,
69°W; Terrizzano et al., 2017), (4) Calingasta Basin (32°S, 69.5°W; Peri et al., 2022), (5) Río Deseado (47°S, 72°W; Tobal et
al., 2021), (6) Río Santa Cruz (50°S, 73°W; Milanez Fernandes, 2023). Inset map of South America indicates Fig. 1A extent
and the location of the Lake Titicaca (square symbol; Fritz et al., 2007), Salar de Uyuni (triangle symbol; Baker et al., 2001)
and Botuverá Cave (diamond symbol; Wang et al., 2007) paleoenvironmental records. Dashed arrows outline the moisture-
bearing low-level airflow patterns for South America which are deflected by the Andean topography. B) Topography of the
Toro Basin (ca. 4000 km², 1500-5900 m asl) from TanDEM-X (12-m resolution) elevation data. Basin outlined by dashed
black line. Upper basin delineated by white-black bordered rectangle (see Fig. 2). Toro alluvial fans and fluvial terraces
outlined by red and orange rectangles, respectively. Basin outlet and start of long profile in Fig. 2 is shown by blue circle. Sa.
– Sierra.

Approximately 30 km upstream of the 100-kyr cut-and fill terraces in the Toro Basin is a suite of well-
preserved alluvial fan surfaces which extend from tributary catchments that drain the Sierra de Pascha
(Fig. 1). There is limited evidence of sediment storage in these tributary catchments en route to the fans.
With an upstream channel length of ~10 km, this fan record may capture geomorphic change linked to



a higher frequency climate forcing than the downstream terraces. The Toro Basin alluvial-channel
system therefore allows us to explore (1) how channel responses to external perturbations may or may
not propagate downstream, and (2) the differences in landscape response to forcing frequency as a
function of channel length when comparing the upper basin alluvial fan deposits with the lower basin
terrace sequence.

To address these aims, we dated the suite of fan surfaces in the upper Toro Basin using *in situ*-[10]Be
cosmogenic radionuclide (CRN) dating. We used our new Toro fan chronostratigraphy in conjunction
with the fluvial terrace record of Tofelde et al. (2017) to further characterise the evolution of the Toro
Basin over the last million years.
## 2. Regional setting

The Toro Basin (24.5°S) is an intermontane basin in the Eastern Cordillera of NW Argentina, located
between the high elevation Puna Plateau to the west and the low elevation Andean foreland to the east
(Fig. 1). The mainly gravel-bedded Río Toro flows predominantly south from the low relief upper
reaches of the basin with thick successions of preserved sediment, which are the focus of this study
(referred to as the upper Toro Basin herein), through a steep bedrock gorge, before draining into the
Cabra Corral reservoir in the Lerma valley (Marrett and Strecker 2000; DeCelles et al., 2011). The
diffuse shifts in channel steepness along its course are characteristic of arid, tectonically active
landscapes with mechanically strong basement rocks (Fig. 2B, C) (Bernard et al., 2019, Zondervan et
al., 2020; Seagren and Schoenbohm, 2021).

**2.1 Geology and tectonic setting**
The upper Toro Basin is confined by three reverse-fault bounded basement ranges: 1) the Cumbres de
Zamaca bounded by the west-dipping Solá Fault in the west, 2) the Sierra de Almagro bounded by the
northwest-dipping San Bernardo fault in the north, and 3) the Sierra de Pascha Ranges and the east-
dipping Gólgota Fault in the east (Marrett and Strecker 2000) (Fig. 1, 2). The Solá fault has been active
since at least the Pliocene, and tectonic deformation from the Miocene to mid-Pleistocene has been
recorded along the San Bernardo and Gólgota faults (Marrett and Strecker 2000). The Gólgota fault
reactivated after ca. 0.98 Ma (Hilley and Strecker 2005).

This study focuses on a suite of fans that emerge from the tributary catchments of the Sierra de Pascha
and are located ~30 km upstream from the cut-and-fill terraces recording 100-kyr climate cyclicity
described by Tofelde et al. (2017). The Pascha Ranges are characterised by meta-sediments of the Late
Proterozoic-Cambrian Puncoviscana Formation and quartzites and shales of the Cambrian Mesón



Group (Schwab and Schafer 1976; García et al., 2013). Long term rock-uplift rates based on structural
reconstructions range between 0.4 and 0.6 mm/yr (Hilley and Strecker 2005).

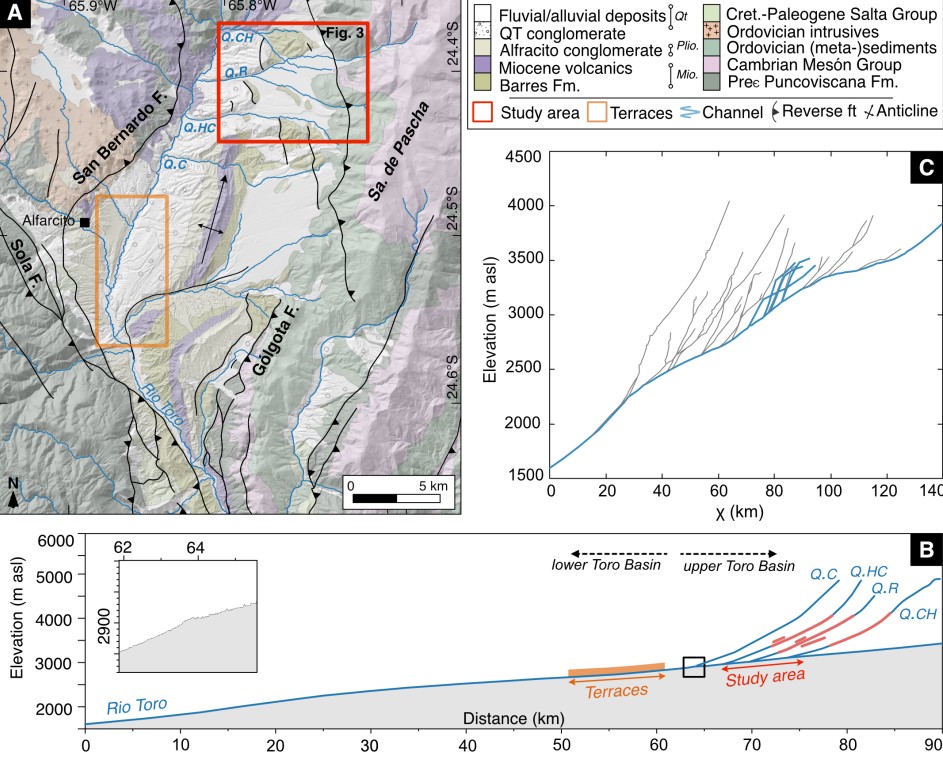

**Figure 2.** Geology and topography of the upper Toro Basin. A) Geologic map with the alluvial fan sequence location (our
study area, Fig. 3) and cut-and-fill terraces described by Tofelde et al. (2017) outlined by red and orange rectangles
respectively. Other terraces extend discontinuously along the basin's channel length but remain undated. Map adapted from
Segemar 250k geological maps and Pingel et al. (2020). Abbreviations: Sa. – Sierra, F – Fault, Q.CH – Quebrada (mountain
stream) Chacra Huaico, Q.R – Quebrada Rosal, Q. HC – Quebrada Huasa Ciénaga, Q.C – Quebrada del Chorro, Q.Ca–
Quebrada Carachi.  B) Long profile of Toro Basin with tributary profiles of upper basin study area. Upper and lower basin
reaches are indicated by dashed arrows. Full basin profile extracted from fluvial network outlined in Fig. 1. Alluvial fan and
terrace surfaces are projected onto profiles. Inset: Higher resolution plot of proposed knickzone at confluence between the Río
Toro and Quebrada del Chorro (outlined in main plot by black box). C) Chi-plot of all channels with a minimum drainage area
of 1 km² within the Toro basin using a reference concavity index of 0.45. Bold lines highlight the main river channel and
tributary catchments within our study area.

The Middle Miocene Barres Sandstone, interbedded with lava flows, and the Pliocene-Pleistocene
Alfarcito Conglomerates are exposed along a north-trending anticline, which lies between the fan
deposits and the Río Toro (Fig. 2A; Mazzuoli et al., 2008; DeCelles et al., 2011; Robledo et al., 2020).
Resistant Barres and Alfarcito units characterise several erosional surfaces that stand ~700 m above the
modern river channel. Incision into these tectonically deformed units by tributaries draining the Sierra
de Pascha is thought to have occurred after 0.98 Ma (Hilley and Strecker, 2005), the age of an
intercalated ash unit dated from the uppermost layers of the Alfarcito Conglomerate (Marrett et al.,



1994). Undeformed Quaternary conglomerates (also called 'Terrace Conglomerates') and
fluvial/alluvial deposits either mantle or infill this tectonically deformed and eroded palaeotopography
(Fig. 2; Marrett and Strecker, 2000; Hilley and Strecker, 2005). The Río Toro sets the local base level
for the Pascha tributaries today (Tofelde et al., 2017).

**2.2 Climatic setting**
Moisture mainly governed by the South American Summer Monsoon (SASM) system is directed by the
South American low-level jet (SALLJ) from the Atlantic Ocean and Amazon Basin to the Central Andes
(Vera et al., 2006; Alonso et al., 2006; Bookhagen and Strecker 2008; Castino et al., 2017). The semi-
arid Toro Basin is located towards the southern limit of this moisture conveyor and receives rainfall that
ranges from ~900 mm/yr at the outlet to < 200 mm/yr in the basin headwaters (Fig. 1; Bookhagen and
Strecker 2008). The Sierra de Pascha acts as an orographic barrier, causing the eastern flanks of the
range to be comparatively wetter than the basin interior. The intensity of the SASM and resultant
moisture supply to the Central Andes has been variable over time (see Baker and Fritz, 2015 for detailed
review). Paleoenvironmental records from Argentina, Chile and Bolivia show that SASM precipitation
has varied with changes in insolation over 19 to 25-kyr (precession) (Godfrey et al., 2003, Fritz et al.,
2004, 2010; Placzek et al., 2006; Bobst et al., 2001) and 100-kyr (eccentricity) (Fritz et al., 2007;
Gosling et al., 2008) cycles. The Central Andes are also subject to increased rainfall during periods of
northern hemispheric cooling, whereby the Atlantic part of the intertropical convergence zone (ITCZ)
is forced southward, bringing moisture with it (Broccoli et al., 2006; Mosblech et al., 2012; Novello et
al., 2017; Crivellari et al., 2018). These cold and wet conditions correlate with phases of glacial advance
and rising lake levels (Haselton et al., 2002; Vizy and Cook, 2007; Martin et al., 2018; Mey et al., 2020).

Successions of glacial moraines are preserved within the Sierra de Pascha tributary catchments and are
indicative of repeated late Quaternary glaciations (Tofelde et al., 2018). Glacial records proximal to the
Toro Basin (24-27.2°S) underline the sensitivity of Andean glaciers to SASM precipitation intensity
and temperature (Martini et al., 2017; Zech et al., 2017; Luna et al., 2018; D'Arcy et al., 2019a; Mey et
al., 2020). The timing of regional glacial stages is invariably in phase with insolation cycles, periods of
SASM strengthening and/or northern hemispheric events (e.g., Younger Dryas, Last Glacial Maximum)
(D'Arcy et al., 2019a).

**2.3 Basin sediment infilling and incision**
Thick successions of sediment, together with subtle knickzones and hairpin turns in the Río Toro reflect
a complex late Cenozoic history of basin filling and evacuation (Strecker et al., 2009; Hain et al., 2011;
Vezzoli et al., 2012; Pingel et al., 2020), base level perturbations and tectonic deformation (Marrett and
Strecker, 2000; Hilley and Strecker, 2005; Tofelde et al., 2017), and drainage reorganization (Seagren



and Schoenbohm, 2021; Seagren et al., 2022). Given our interest in the Quaternary deposits of the upper
Toro Basin, we focus our attention on how the basin has evolved over the last one million years.

After deposition of the Alfarcito conglomerates concluded at ca. 0.98 Ma, the Toro Basin was evacuated
to a base level lower than today (Hilley and Strecker, 2005). Renewed hydrological connectivity
between the Toro Basin and the Lerma Valley likely caused widespread basin sediment evacuation and
incision of the (paleo)topography. Uplift of the Sierra de Pascha Sur also recommenced sometime after
ca. 0.98 Ma (Hilley and Strecker, 2005). The newly uplifted range impeded the delivery of precipitation
to the basin interior, and by ca. 0.8 Ma, the semi-arid conditions of today were established (Kleinert and
Strecker 2001; Strecker et al. 2007; Pingel et al., 2020). The mechanically strong basement rocks, and
a potentially reduced sediment transport capacity, meant that incision was unable to keep pace with the
renewed rock uplift. This forced widespread aggradation and a decrease in relief upstream of the
Gólgota fault, and channel steepening within the bedrock gorge cutting through the Sierra de Pascha
Sur (Fig. 2; Hilley and Strecker, 2005; Strecker et al., 2009; García et al., 2013). External drainage
either became restricted or ceased at this time (Marrett et al. 1994; Hain et al., 2011; Pingel et al.,
2019a). Evidence for a similar sequence of events is seen in the Humahuaca, Casa Grande and
Calchaquí basins (23°S), where renewed range uplift reduced hydrological connectivity and caused
sediment infilling (Robinson et al., 2005; Hain et al., 2011; García et al., 2013; Pingel et al., 2013, 2016,
2019a; Streit et al., 2017; Seagren et al., 2022). Although there are some uncertainties about the exact
timing, connectivity between the Toro Basin and the foreland is thought to have been re-established due
to external base-level change (Seagren and Schoenbohm, 2021).

The Quaternary "Terrace Conglomerates" were deposited within the Toro Basin starting from ca. 0.94
Ma and are considered part of this phase of uplift-induced basin infilling (Hilley and Strecker, 2005). A
flight of six fluvial terrace levels in the lower basin are preserved between 20 and 200 m above the
modern Río Toro (Fig. 2). Cosmogenic exposure-age dating of terraces, burial dating of the sediments,
and zircon U-Pb ages of intercalated ashes from the terrace levels revealed multiple 100-kyr cut-and-
fill sedimentary cycles starting from ca. 500 ka (Tofelde et al., 2017). The phases of incision correspond
with cold, wet glacial periods, when sediment transport capacity apparently exceeded sediment flux,
whereas aggradation occurred when sediment transport was considerably reduced (Tofelde et al., 2017).
Moreover, the calculated net incision rate through the terrace sequence of 0.4 mm/yr from ca. 500 ka is
consistent with long term rock-uplift rates of the Sierra de Pascha Sur at the basin outlet (Hilley and
Strecker, 2005). Tofelde et al. (2017) thus concluded that while the renewed uplift of the Sierra de
Pascha Sur helped initiate the deposition of the Terrace Conglomerates, the periodicity of the cut-and-
fill cycles is best explained by orbitally driven climate forcing, with net incision likely associated with
the channel response to ongoing rock-uplift. Today, catchment-averaged erosion rates for catchments
draining the Sierra de Pascha range between <0.03 and 0.12 mm/yr (Tofelde et al., 2018).




## 3.  Methodology

To evaluate past channel behaviour and landscape response to climate and/or tectonic forcing for the
upper Toro Basin, we applied CRN exposure dating to the suite of fan surfaces along the western front
of the Sierra de Pascha (Fig. 1, 2).

Alluvial fan CRN ages record the timing of active sediment deposition or surface stability between
periods of channel avulsion and incision (Dühnforth et al., 2007; D'Arcy et al., 2019b), which lead to
abandonment of the fan surface. This abandonment can occur due to changes in sediment supply
(Brooke et al., 2018; Tofelde et al., 2019), tectonic deformation and base level change (Ganev et al.,
2010; Mouslopoulou et al., 2017), climate-induced changes in water discharge (Steffen et al., 2010;
Savi et al., 2016) or drainage reorganization (Bufe et al., 2017). Because fan surfaces can remain active
for $10^{2\text{-}5}$ years before being incised (Cesta and Ward, 2016; Dühnforth et al., 2017; Ratnayaka et al.,
2019; Peri et al., 2022), the age distribution or minimum exposure age of boulders on an alluvial fan
surface will not necessarily tightly constrain the timing of abandonment. Instead, the distribution of
CRN ages, after excluding clear outliers, more likely reflects phases of fan activity, and at best, provide
a minimum age limit for the onset of incision leading to eventual surface abandonment (D'Arcy et al.,
2019b).

We mapped the upper Toro Basin fans using TanDEM-X (12 m-resolution) data and Google Earth
imagery. The stratigraphic relationships among the different fan surfaces were used to inform the
cosmogenic radionuclide (CRN) sampling strategy (e.g., McFadden et al., 1989; Hughes et al., 2010;
Hedrick et al., 2013).

Supporting topographic, fan and channel data were extracted from the DEM using TopoToolbox
functions in MATLAB (Schwanghart and Scherler, 2014) and geospatial toolboxes (GRASS, GDAL)
in QGIS. We also compiled a set of climate (Berger and Loutre, 1991; Baker et al., 2001; Imbrie et al.,
2006; Fritz et al., 2007; Wang et al., 2007; Lisiecki and Raymo, 2009), paleoenvironmental (Hilley and
Strecker, 2005; Tofelde et al., 2017; Pingel et al., 2020), glacial (Martini et al., 2017; Zech et al., 2017;
Luna et al., 2018; D'Arcy et al., 2019a; Mey et al., 2020) and geomorphic (Terrizzano et al., 2017;
Tofelde et al., 2017; Pingel et al., 2019b; Tobal et al., 2021; Peri et al., 2022; Milanez Fernandes, 2023)
records for the Andes to help contextualise our results.

### 3.1 CRN dating
We collected a total of 30 quartzite boulder surface samples from eight fan surfaces (Fig. 3). Between
three and four boulders were sampled per surface. Each surface was named 'Qf' for 'Quaternary fan',





followed by a number which referred to its stratigraphic position. For example, Qf_1 sits ~200 m above
the modern river channel, and as the highest elevation surface of the study area, it was anticipated to be
the oldest fan.

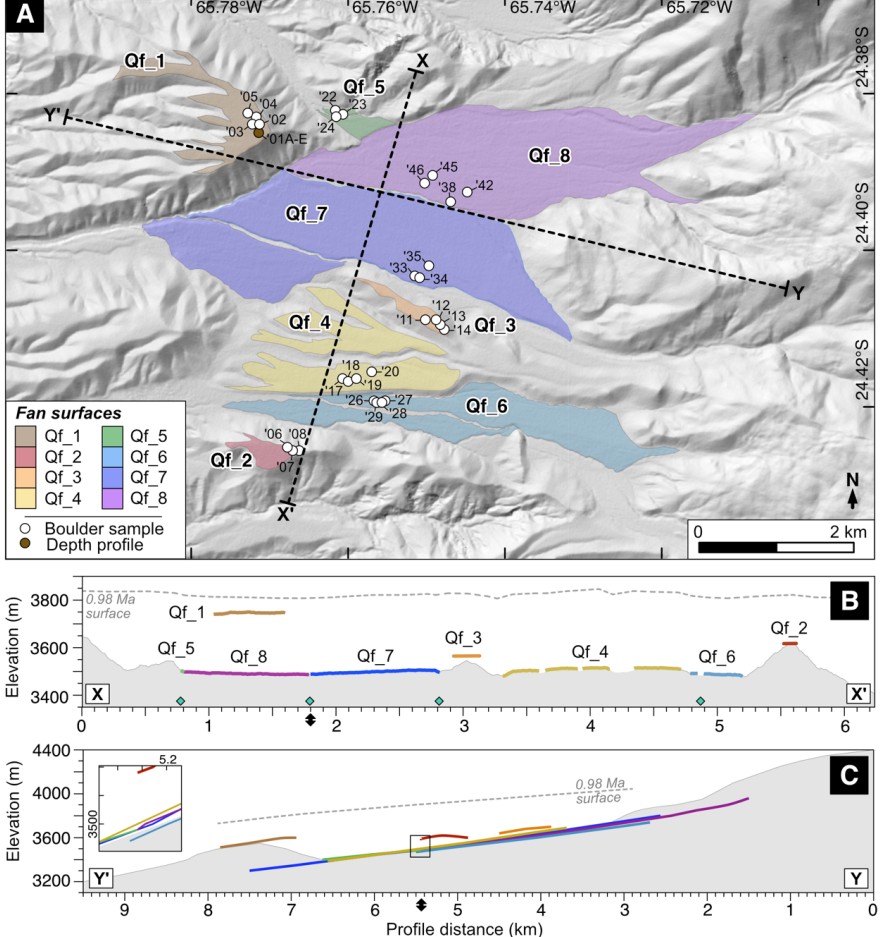


**Figure 3.** Alluvial fan surfaces of the upper basin. A) Hillshade map of the dated fan surfaces with boulder and depth profile
sampling locations shown. Sample names have been abbreviated (e.g.: TB19_05: '05). X-X' and Y-Y' projection lines of Fig.
3B and 3C are represented by dashed black lines. B) Fan sequence stratigraphy shown by fan surfaces projected onto X-X'.
Qf_2 and Qf_3 surface widths are slightly exaggerated to improve visibility. Modern topography shaded in grey. The 0.98Ma
surface (grey dashed line) is modelled from sediment evacuation estimates of Hilley and Strecker (2005). Location of active
fluvial channels indicated by green diamond symbol. C) Fan surfaces projected onto Y-Y'. Inset plot provides higher resolution
view of projections (outlined by black rectangle). Projection line intersection is indicated by black double arrow.


Each sampled boulder was embedded within the fan surface, located away from channels, and within
the distal zone of the landform. This sampling strategy reduced the likelihood that the boulders were
sourced from adjacent hillslopes or were part of a depositional event following landform abandonment
(D'Arcy et al., 2019b; Orr et al., 2021). The sampled boulders were the largest, freshest boulders that



we were able to identify within the distal zone. However, we cannot definitively discount the possibility
that the boulders experienced some weathering, surface spallation or fracturing in the past.

We removed between 400 and 1000 g of sample from the upper three centimetres of each boulder
surface. The samples were crushed and then sieved to isolate the 250–500 µm grainsize fraction needed
for CRN dating. Sample cleaning, purification, extraction and oxidation of $^{10}$Be, and target preparation
for AMS measurement was conducted in the Helmholtz Laboratory for the Geochemistry of the Earth
Surface (HELGES) at the German Research Centre for Geosciences (GFZ-Potsdam) using the
procedures outlined by von Blanckenburg, (2004) and Wittmann et al. (2016). AMS measurements were
completed at the Cologne AMS facility at the University of Cologne, Germany.

Exposure ages derived from *in situ* produced $^{10}$Be concentrations were calculated using the CREp online
calculator (Martin et al., 2017) with the regional reference (SLHL) production rate of 3.74 (±0.09) at/
g yr for the high-elevation (> 3400 m asl) Central Andes (Blard et al., 2013; Kelly et al., 2015; Martin
et al., 2015), and the LSD scaling scheme (Lifton et al., 2014). Further information about the boulder
samples, the CRN laboratory procedure, blank ratios, and age calculation is provided in Supplement 1
and 2.

The probabilistic model for inferring the timing of fan surface abandonment from D'Arcy et al. (2019
b) was applied to fans with exposure ages of less than ca. 300 ka. The model uses the exposure ages of
boulders on the fan surface to generate a probability distribution of abandonment ages and a most
probable abandonment age. The model was not applied to older fan surfaces, which have large age
distributions (>100 kyr range) and likely have some inheritance and/or surface erosion (Phillips et al.,
1990; Tobal et al., 2021). Working with chronological data at this coarse resolution over $10^{5-6}$-year
timescales means that even the most sophisticated inheritance/erosion models are limited in their ability
to estimate the timing of landform abandonment (e.g., Prush and Oskin, 2020; Dortch et al., 2022). For
the Toro fans where this applies, we use the age distribution, stratigraphic order of the fans, and youngest
exposure age as a guide for the timing of abandonment.

**3.2 $^{10}$Be depth profile**
To help substantiate our new $^{10}$Be boulder dataset we also resampled the Qf_1 $^{10}$Be depth profile,
referred to as P6b by Tofelde et al. (2017), and corresponding to their terrace level T6.  The original
profile was limited to five samples, which were sampled over relatively broad depth intervals (0–10cm,
18–28 cm, 25–81 cm, 82–164 cm, 164–210 cm). To obtain more highly resolved $^{10}$Be data for this
surface, particularly in the upper 100 cm, five samples of > 65 pebbles each were extracted from the
following depth intervals (cm): 0–10, 20–30, 40–50, 60–70 and 115–125. The pebble samples were



crushed and sieved, and the 500–1000 µm fraction was reserved for CRN dating. Subsequent laboratory
procedures followed that of the boulder samples.

The Qf_1 $^{10}$Be depth profile, using combined $^{10}$Be data from this study and from Tofelde et al., (2017),
was used to determine an exposure age using the Hidy et al. (2010) Monte Carlo simulator. Further
details are provided in Supplement 1 and 2.

**4.  Results**

We use the upper Toro Basin alluvial fan elevations, surface characteristics, and CRN ages to identify
two generations of fan surfaces. The studied fans are predominantly matrix-supported conglomerates
with sub-angular to rounded pebble and cobble clasts. Weathered desert pavements cap many of the fan
surfaces; a layer of finer sands and gravels are overlain by pebbles, cobbles, and boulders (e.g.,
McFadden et al., 1989; Tofelde et al., 2017).

The Generation 1 (G1) fan surfaces, comprising Qf_1 through 4, are stratigraphically the highest in the
record and are positioned ~200 to 50 m above the modern river channel(s) (Fig. 3). The fan surfaces are
moderately to highly weathered, with some evidence of surface boulder spallation (Fig. 4). With a few
rare exceptions, the G1 sampled boulders are smaller than those sampled from the lower Generation 2
(G2) surfaces. The G1 and G2 boulders have b-axis lengths which range from 30 to 80 cm and 30 to
140 cm, respectively (Supplement 2). The CRN exposure ages from the G1 surfaces range between ca.
970 and 340 ka (Table 1; Fig. 5, 6).

G2 is comprised of fans Qf_5 through 8, which have surfaces within 10 m elevation of the modern
channel(s) (Fig. 3). These moderately weathered surfaces retain debris flow deposits, evidence of past
channel avulsion and sparse human infrastructure (e.g., stone walls). The CRN exposure ages of this
younger fan generation range between ca. 100 and 20 ka, with estimated surface abandonment ages
after ca. 70 ka (Table 1; Fig. 7).













**Table 1.** Sample properties, measured [10]Be concentrations and calculated exposure ages of each sampled boulder from the Toro fans. Further sample and age calculation details are provided in the Supplement 2 and 3.

| Sample | Location | | | Sample thickness | Shielding correction | Be-10 concentration | | Be-10 exposure ages[1] | |
|---|---|---|---|---|---|---|---|---|---|
| | Latitude | Longitude | Elevation | | | Concentration | Uncertainty | Age | Uncertainty |
| | (°S) | (°W) | (m asl) | (cm) | | ($10^6$ at/g $SiO_2$) | ($10^6$ at/g $SiO_2$) | (ka) | (ka) |
| **Qf_1** | | | | | | | | | |
| TB19_02 | -24.38492 | -65.76890 | 3556 | 1 | 0.990 | 24.20 | 0.78 | 966.63 | 109.78 |
| TB19_03 | -24.38492 | -65.76890 | 3556 | 1 | 0.990 | 16.02 | 0.52 | 593.11 | 59.10 |
| TB19_04 | -24.38492 | -65.76890 | 3556 | 1 | 0.990 | 22.33 | 0.72 | 884.41 | 95.34 |
| TB19_05 | -24.38492 | -65.76890 | 3556 | 1 | 0.990 | 16.97 | 0.55 | 639.17 | 63.94 |
| **Qf_2** | | | | | | | | | |
| TB19_06 | -24.42522 | -65.76775 | 3560 | 1 | 0.999 | 11.36 | 0.37 | 391.94 | 37.91 |
| TB19_07 | -24.42566 | -65.76682 | 3570 | 2 | 0.999 | 17.00 | 0.55 | 631.77 | 64.10 |
| TB19_08 | -24.42568 | -65.76607 | 3581 | 2 | 0.999 | 10.18 | 0.33 | 336.94 | 33.17 |
| **Qf_3** | | | | | | | | | |
| TB19_11 | -24.40882 | -65.75023 | 3644 | 1 | 0.998 | 15.45 | 0.50 | 533.56 | 52.88 |
| TB19_12 | -24.40918 | -65.74864 | 3658 | 3 | 0.998 | 18.06 | 0.59 | 651.82 | 66.21 |
| TB19_13 | -24.40976 | -65.74810 | 3660 | 3 | 0.998 | 17.77 | 0.58 | 634.67 | 64.63 |
| TB19_14 | -24.41011 | -65.74773 | 3673 | 3 | 0.998 | 11.18 | 0.37 | 361.38 | 35.49 |
| **Qf_4** | | | | | | | | | |
| TB19_17 | -24.41665 | -65.76059 | 3509 | 1 | 0.999 | 14.73 | 0.48 | 548.44 | 54.60 |
| TB19_18 | -24.41675 | -65.76000 | 3512 | 2 | 0.999 | 17.26 | 0.56 | 679.67 | 68.23 |
| TB19_19 | -24.41654 | -65.75923 | 3519 | 3 | 0.999 | 19.06 | 0.61 | 778.81 | 79.15 |
| TB19_20 | -24.41533 | -65.75681 | 3541 | 1 | 0.999 | 21.41 | 0.69 | 847.34 | 90.30 |
| **Qf_5** | | | | | | | | | |
| TB19_22 | -24.38245 | -65.76145 | 3404 | 2 | 0.990 | 2.02 | 0.07 | 70.63 | 6.28 |
| TB19_23 | -24.38263 | -65.76109 | 3407 | 2 | 0.995 | 2.34 | 0.08 | 82.69 | 7.35 |



| | | | | | | | | | |
|---|---|---|---|---|---|---|---|---|---|
| TB19_24 | -24.38275 | -65.76144 | 3405 | 3 | 0.995 | 2.77 | 0.09 | 98.81 | 8.82 |
| **Qf_6** | | | | | | | | | |
| TB19_26 | -24.41923 | -65.75623 | 3531 | 2 | 0.998 | 2.16 | 0.07 | 69.97 | 6.27 |
| TB19_27 | -24.41921 | -65.75578 | 3532 | 1 | 0.998 | 2.52 | 0.08 | 81.85 | 7.31 |
| TB19_28 | -24.41924 | -65.75569 | 3541 | 2 | 0.998 | 2.22 | 0.08 | 71.11 | 6.46 |
| TB19_29 | -24.41941 | -65.75652 | 3525 | 3 | 0.998 | 2.47 | 0.08 | 82.00 | 7.33 |
| **Qf_7** | | | | | | | | | |
| TB19_33 | -24.40346 | -65.75108 | 3557 | 1 | 0.998 | 1.22 | 0.04 | 38.78 | 3.46 |
| TB19_34 | -24.40371 | -65.75107 | 3555 | 2 | 0.998 | 1.87 | 0.06 | 59.28 | 5.36 |
| TB19_35 | -24.40203 | -65.74977 | 3563 | 3 | 0.998 | 2.11 | 0.07 | 66.94 | 6.13 |
| **Qf_8** | | | | | | | | | |
| TB19_38 | -24.39402 | -65.74711 | 3533 | 1 | 0.997 | 1.43 | 0.05 | 44.34 | 4.08 |
| TB19_42 | -24.39275 | -65.74500 | 3553 | 1 | 0.997 | 1.43 | 0.05 | 43.65 | 4.04 |
| TB19_45 | -24.39043 | -65.74940 | 3510 | 1 | 0.997 | 0.63 | 0.02 | 22.37 | 1.85 |
| TB19_46 | -24.39140 | -65.75027 | 3502 | 1 | 0.997 | 1.44 | 0.05 | 45.32 | 4.212 |

1: LSD scaling scheme (Lifton et al., 2014), ERA40 Atmosphere Model (Uppala et al., 2005), LSD framework for geomagnetic correction (Lifton et al., 2014), Reference (SLHL) production rate: 3.74+0.09 at/g/yr. Sample density: 2.75 g cm$^{-3}$. Erosion: 0 mm yr$^{-1}$





377

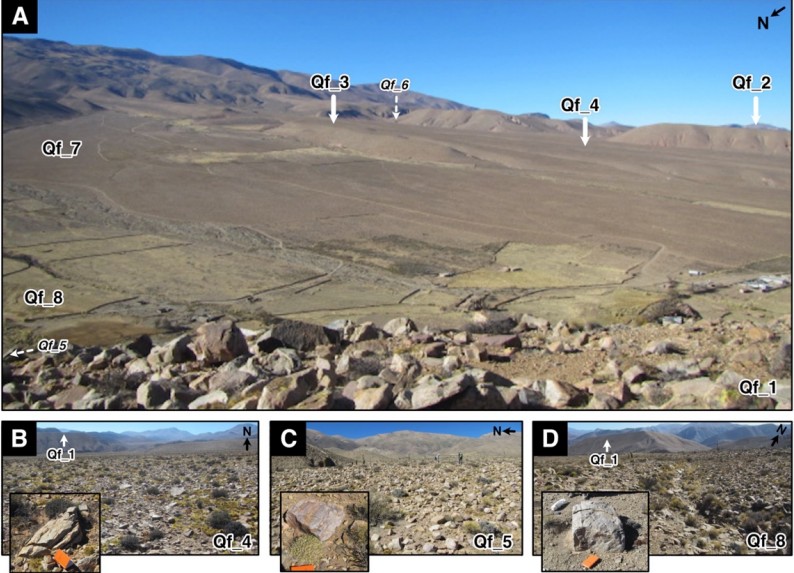

378

**Figure 4.** Images of the alluvial fan sequence of the upper Toro Basin. A) Image taken from Qf_1 surface (facing SE) with
fan surfaces labelled. Italicized text with arrows indicates location of surfaces that are not clearly in shot. B) Qf_4 surface.
Inset image of sampled boulder TB19_19. C) Qf_5 surface. Inset image of sampled boulder TB19_22. D) Qf_8 surface. Inset
image of sampled boulder TB19_44. Images B–D encompass full age range of sampled surfaces. Further images of the fan
surfaces and $^{10}$Be samples are provided in Supplement 3.

## 4.1 Generation 1

Qf_1 is the highest fan surface of the record (~200 m above the modern channel), which extends from
the Quebrada Rosal tributary catchment. The fan comprises part of the Quaternary conglomerates,
which overlie the Barres Sandstone Formation (Fig. 2, 3). The depth profile is composed of four
sedimentary units that coarsen with depth: silts and fine sands (0–20 cm), fine-coarse sand (20–60 cm),
coarse sand and gravel (60–180 cm) and gravels (>180 cm). Consistent with the original profile, the
new $^{10}$Be sample concentrations decrease exponentially with depth (Fig. 5; Table 2). Qf_1 has a
Bayesian most-probable exposure age of 715.8 $^{+35}/_{-217}$ ka (2σ upper age: 750.8 ka, 2σ lower age: 498.8
ka) and 0.26±0.42x10$^6$ atoms/g of inheritance. Within the simulator, we constrained fan surface erosion
and inflation by setting the erosion rate to range between -0.02 and 0.2 cm/ka and using maximum and
minimum erosion thresholds of -10 and 50 cm, respectively. While this modelled exposure age is
consistent with the age estimated earlier by Tofelde et al. (2017) of 732 $^{+53}/_{-56}$ ka assuming a stable
surface, or 644 $^{+43}/_{-49}$ ka accounting for surface inflation, Tofelde et al. (2017) preferred the exposure
age they derived from surface pebbles of 453 ± 33 ka.

The exposure ages of boulder samples TB19_03 and TB19_05 are in agreement with the depth profile
results, yielding exposure ages of 639.17 ± 63.94 and 593.11 ± 59.10 ka (2σ uncertainty). The two

49152



remaining boulders (TB19_02, TB19_04) yielded older exposure ages of 966.63 ± 109.78 and 884.41
± 95.34 ka.

**Table 2.** Sample depths and measured [10]Be concentrations of Qf_1 depth profile. Fan age calculated with the Hidy
et al. (2010) Monte Carlo depth profile simulator was 715.8 $^{+35}$/$_{-217}$ ka. Inheritance measured: 0.26±0.42 x10[6] at/g.

| Sample[1] | Sample depth | | Be-10 concentration | |
|---|---|---|---|---|
| | Depth *(cm)* | Uncertainty *(cm)* | Concentration *(10[6] at/g SiO$_2$)* | Uncertainty *(10[6] at/g SiO$_2$)* |
| BBC-0 | 5 | 5 | 14.70 | 0.18 |
| TB19_01A | 5 | 5 | 14.97 | 0.48 |
| BBC-1 | 23 | 5 | 11.80 | 0.11 |
| TB19_01B | 25 | 5 | 12.14 | 0.39 |
| TB19_01C | 45 | 5 | 10.88 | 0.35 |
| BBC-2 | 53 | 28 | 7.76 | 0.07 |
| TB19_01D | 65 | 5 | 8.76 | 0.28 |
| TB19_01E | 120 | 5 | 4.94 | 0.16 |
| BBC-3 | 123 | 41 | 5.21 | 0.06 |
| BBC-4 | 187 | 23 | 2.30 | 0.03 |

1: TB19_01A-E from this study. 'BBC-1-4' from Tofelde et al. (2017).

Surface Qf_2, the second highest surface (ca. 130 m above the closest modern channel), also overlies
the Barres Sandstone and likely extends from the Quebrada Huasa Ciénaga and Quebrada del Chorro
catchments. CRN exposure ages from three boulders range from 631.88 ± 64.10 to 336.94 ± 33.17 ka.

The Qf_3 surface is positioned ca. 60 m above the closest modern channel and extends from the
Quebrada Rosal tributary catchment. The surface yields three CRN boulder exposure ages that cluster
between 651.82 ± 66.21 and 533.56 ± 52.88 ka, and one younger age of 361.38 ± 35.49 ka.

Qf_4 has a highly dissected fan surface which is the lowest stratigraphically of the G1 fans; the fan is
positioned ca. 40 m below the Qf_3 surface and ca. 30 m elevation above the modern channel. Four
boulder exposure ages range from 911.61 ± 100.27 to 548.44 ± 54.60 ka.

**4.2 Generation 2**
Qf_5 is a small G2 surface that sits ca. 10 m above the neighboring Qf_8 fan. Qf_5 has three exposure
ages that range from 98.81 ± 8.82 to 70.63 ± 6.28ka, with a most probable abandonment age of 61.8
$^{+13.5}$/$_{-33.6}$ ka (no ages excluded as outliers).

Qf_6's surface is characterized by moderately weathered debris flow deposits with clusters and
elongated ridges of boulders. Exposure ages range between 82.00 ± 7.33 and 69.97 ± 6.27 ka from the



four boulders, with an estimated surface abandonment age of 66.2 $^{+11.0}/_{-17.5}$ ka (no ages excluded as
outliers).

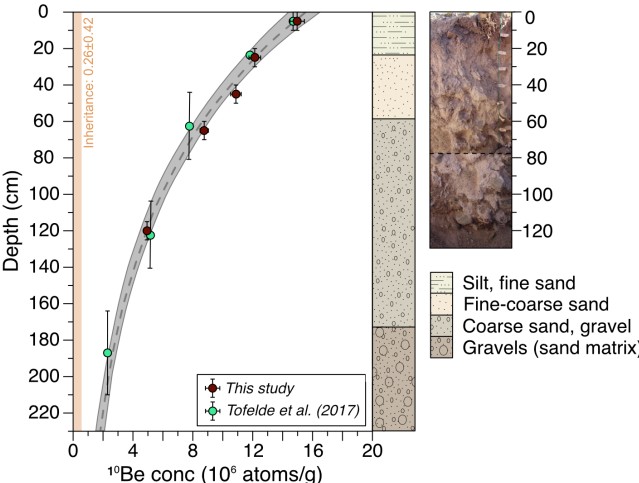


**Figure 5.** $^{10}$Be concentration with depth for Qf_1 profile alongside sedimentary log and stitched field image of the profile pit.
Each sample was collected over a depth range represented by a vertical error bar. Horizontal error bar represents the 1σ
analytical uncertainty for the nuclide concentration. The Hidy et al. (2010) Monte Carlo simulator fit 100,000 curves (grey
shading) to profile and generated most probable fit (grey dashed line). Modelled inheritance is shown by orange line. *Profile
6b data, rather than 6a, from the supplementary materials is used in simulation, due to the mislabelling of the profile in Fig. 4
of Tofelde et al. (2017).

Despite Qf_7 being located within 5 m elevation of the youngest G2 fan Qf_8, this large fan appears
more weathered than Qf_8. Qf_7 has CRN exposure ages of 66.94 ± 6.13, 59.28 ± 5.35 and 38.78 ±
3.47 ka. The surface abandonment ages including and excluding the youngest age are 33.9 $^{+7.4}/_{-25.1}$ and
52.9 $^{+11.0}/_{-16.3}$ ka, respectively.

Surface Qf_8 yielded a cluster of older ages that range between 45.32 ± 4.2 and 43.65 ± 4.04 ka and a
single younger age of 22.37 ± 1.83 ka. Abandonment ages including and excluding the youngest age
are 19.4 $^{+4.1}/_{-19.4}$ and 42.4 $^{+6.5}/_{-7.5}$ ka, respectively. The surface is covered with relatively unweathered
debris flow deposits and large varnish-free boulders.

**5.  Discussion**

While there are some nuances to the Toro Basin fan record, our new CRN dataset enables us to identify
significant phases of net incision since ca. 0.98 Ma, capture the channel response to external forcing
over a range of timescales and cyclicities, and gain further insight into the late Quaternary evolution of
the Toro Basin.



**5.1 Timing of alluvial fan development and abandonment**

CRN age uncertainties on the order of $10^{4-5}$ years and a wide range of fan exposure age distributions on individual surfaces present some challenges when interpreting the Toro fan chronostratigraphy, which is crucial for comparison with potential external forcing conditions. Constraining the geological uncertainties of the CRN ages, particularly for old fan surfaces, is often challenging (Owen et al., 2014). For this reason, we use geological, topographic and paleoenvironmental data alongside the $^{10}$Be data to interpret the alluvial fan record. The coarse resolution of the G1 $^{10}$Be record means that while we can reflect upon long term shifts in channel behaviour for the upper Toro Basin, we must exercise caution when linking this record to specific forcing or events (Gray et al., 2014; Dühnforth et al., 2017; Orr et al., 2021). Pairing the $^{10}$Be record with cosmogenic $^{21}$Ne in the future may help to decipher some of the complexities in the exposure histories of the boulders; $^{21}$Ne is well suited for quantifying long term landscape change in arid, low erosion environments (Dunai et al., 2005; Ma and Stuart, 2018).

*5.1.1 Fan Generation 1*

The ~200-m elevation difference between the highest and lowest fan surface among Generation 1 means that the G1 surfaces could not have been active simultaneously (Fig. 6). Substantial inheritance and/or erosion has therefore likely affected individual boulders from these surfaces and offers one explanation for the broad spread in ages (>400 kyr) for each.

Pairing the Qf_1 $^{10}$Be depth profile with the surface boulder exposure ages means that we can more robustly constrain the oldest phase of fan development within the study area and use it as a benchmark when evaluating the remainder of the G1 fan record. The most recent phase of Qf_1 surface activity and/or stability is constrained by the depth profile data and two boulders to between ca. 750 and 600 ka. In this case, we believe that CRN inheritance may explain why the remaining two boulders (TB19_02, TB19_04) from this surface yield exposure ages that exceed ca. 800 ka. Considering the whole suite of boulder ages for the G1 fans, which mostly exceed 500 ka, we find it unlikely that the age of 453 ± 33 ka (based on surface pebbles) originally reported by Tofelde et al. (2017) for Qf_1 is correct.

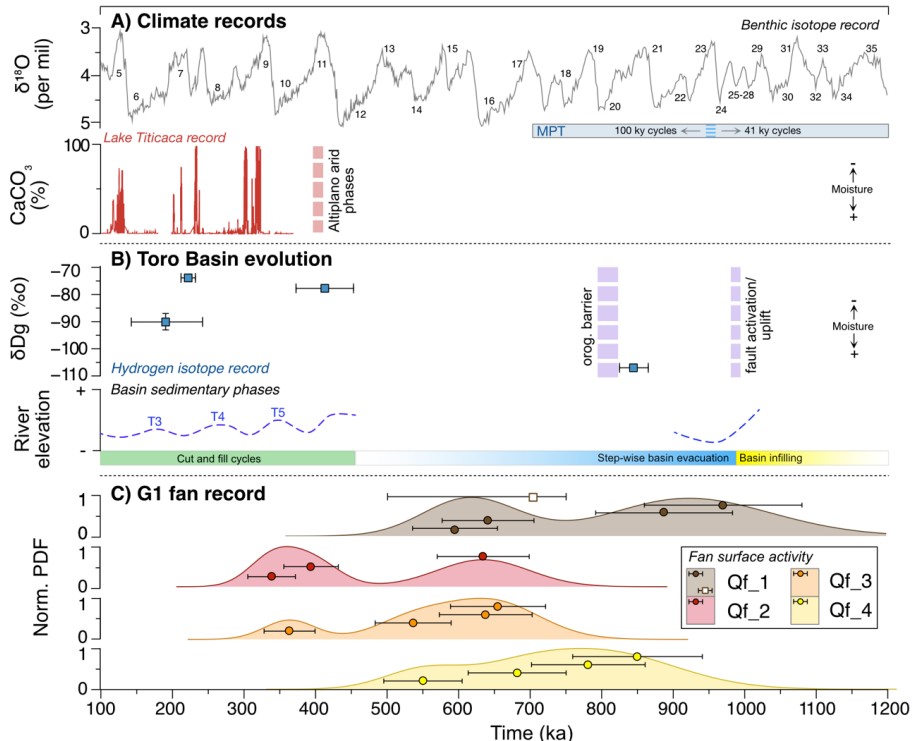

**Figure 6.** Comparison between the G1 fan $^{10}$Be dataset and records of Toro Basin evolution and climate. A) Benthic isotope record (Lisiecki and Raymo 2009) displayed alongside Marine Isotope Stages (MIS) and Mid-Pleistocene Transition labelling and the Lake Titicaca sediment core record (CaCO$_3$ concentration) from Fritz et al. (2007). B) Toro basin evolution. Climatic variability represented by hydrogen isotope record of Pingel et al. (2020). Basin sedimentary and tectonic phases plotted with respect to inferred river elevation over time, as observed by this study and described by Hilley and Strecker (2005), Tofelde et al. (2017) and Pingel et al. (2020). Fluvial terrace record (T3-6) from Tofelde et al. (2017). C) $^{10}$Be surface boulder ages and normalised probability density functions of the G1 surfaces. Horizontal error bars represent the 1σ uncertainty for the exposure ages. Bayesian modelled surface age of Qf_1 (715.8 $^{+35}/_{-217}$ ka) derived from depth profile (Fig. 5) is denoted by square point.

Given the stratigraphic positions of Qf_2 and Qf_3, we think that it is unlikely that active streams were present on these surfaces after ca. 400 ka. For this reason, we suggest that the younger ages for these surfaces are the result of erosion. These surfaces also must be older than surface Qf_4, which yielded a youngest age of ca. 550 ka.

Inheritance also likely explains the old (>750 ka) boulders on Qf_4, which is stratigraphically younger than Qf_1 and cannot have been active at the same time.

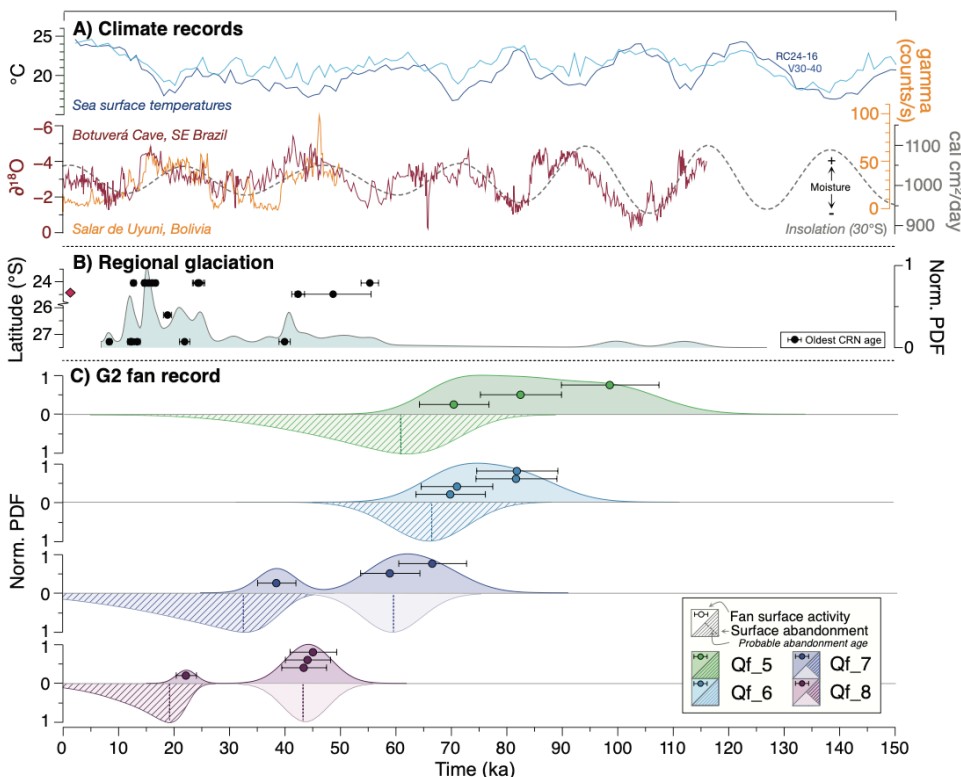

**Figure 7.** Comparison between the G2 fan $^{10}$Be dataset and regional climate and glacial records. A) Climate records. Sea surface temperatures from Imbrie et al. (2006), insolation from Berger and Loutre (1991), Botuverá Cave, SE Brazil speleothem record from Wang et al. (2007) and Salar de Uyuni, Bolivia lake record from Baker et al. (2001). B) CRN glacial chronologies from the Central Andes : Nevado de Chañi (24°S, 65.7°W, Martini et al., 2017, Mey et al., 2020), Quevar Volcano (24.4°S, 66.8°W, Luna et al., 2018), Sierra de Quilmes (26.2°S, 66.2°W, Zech et al., 2017) and the Sierra Aconquija (27.2°S, 66.1°W, D'Arcy et al., 2019a). Location of Toro Basin (24.4°S, 66.7°W) is indicated by red diamond symbol. C) $^{10}$Be surface boulder ages and normalised probability density functions of the G2 surfaces. Horizontal error bars represent the 1σ uncertainty for the exposure ages. Normalised PDF of fan surface abandonment (hashed shading) calculated using the D'Arcy et al. (2019b) probabilistic model for fan surface abandonment. Surface abandonment for Qf_7 and Qf_8 without youngest boulder ages (TB19_33 and TB19_45, respectively) shown by PDFs with opaque solid shading. Most probable abandonment ages denoted with dashed vertical lines- Qf_5: 61.8 $^{+13.5}/_{-33.6}$ ka, Qf_6: 66.2 $^{+11.0}/_{-17.5}$ ka, Qf_7: ca. 33.9 $^{+7.4}/_{-25.1}$ ka (52.9 $^{+11.0}/_{-16.3}$ ka), Qf_8: 19.4 $^{+4.1}/_{-19.4}$ ka (42.4 $^{+6.5}/_{-7.5}$ ka).

Given these complexities in the fan chronostratigraphy, rather than identifying discrete phases of aggradation and incision for each fan surface, we suggest that the G1 fan record can instead be used to capture an extended phase of net incision within the Sierra de Pascha tributaries. By comparing the G1 fan record with the modelled palaeotopography of Hilley and Strecker (2005), we estimate that ~100 m of net incision (~0.01 mm/yr) occurred within the upper basin between ca. 0.98 Ma and 800 ka, at which point the Qf_1 surface became active (Fig. 3B, C, Fig. 8). Approximately 200 m of net incision (~0.07 mm/yr) then followed between ca. 800 ka and the complete abandonment of the G1 fans by ca. 500 ka (when adjusting for age outliers) (Fig. 6), which signals the significant stepwise evacuation of sediment from the upper Toro Basin at this time.





*5.1.2 Fan Generation 2*

The G2 record reveals that after a hiatus in the geomorphic record ca. 500 and 100 ka, fan aggradation and incision is recorded throughout several of the Sierra de Pascha tributaries (Fig. 8). Rather than recording continuous fan activity since ca. 110ka, the distribution of ages for G2 instead likely captures multiple distinct phases of deposition. The G2 fan surfaces have much tighter constrained age distributions (ca. 21 to 40 kyr) compared to the G1 fans, with two G2 fans showing what may be young outliers; the boulders are therefore less likely to be affected by inheritance, but the young outliers may be affected by erosion or tilting by human or animal activity.

**5.2 Drivers of alluvial channel system change and fan/terrace formation**

Before we can explore some of the possible explanations for the alluvial system change recorded in the Toro Basin, we must first consider the specific local conditions needed to help explain the G1 (ca. 800 to 500 ka) and G2 (ca. 100 to 20 ka) fan generations in the upper basin, as well as the fluvial terrace sequence (ca. 370 ka to <75 ka) in the lower basin. Changes in water or sediment discharge, governed by climate, can affect channel slopes and prompt adjustments to the channel bed elevations through incision or aggradation (Howard, 1982; Wickert and Schildgen, 2019; Tofelde et al., 2019). Nevertheless, net incision is essential to preserving the geomorphic record of aggradation-incision cycles. Otherwise, subsequent aggradational phases are likely to bury earlier landforms. Net incision can occur through the channel response to ongoing rock uplift or base level fall (Simpson and Castelltort, 2012), the latter of which may include renewed incision through an aggraded sequence of sediment downstream. While autogenic processes, such as channel avulsion and meander cut-offs may also play a role in channel incision and the formation of discrete fan lobes or terraces (Nicholas and Quine, 2007; Ventra and Nichols, 2014), we consider the scale of channel incision associated with the features of interest (ranging from ca. 10 to hundreds of meters) is beyond the scope of purely autogenic behavior. Below, we consider how climate-modulated changes to water and sediment discharge, together with events that can drive net incision, may have helped to generate, and preserve multiple generations of fans and terraces within the Toro Basin.

*5.2.1 Fan formation from ca. 800 to 500 ka*

The development, entrenchment, and eventual abandonment of the G1 fans could be part of the landscape response to enhanced rock-uplift of the Sierra de Pascha Sur, starting no later than ca. 800 ka (Fig. 8) (Clarke et al., 2010; Mather et al., 2017; Mouchené et al., 2017). However, another mechanism is likely at play because the averages rates of incision between ca. 800 and 500 ka (0.8 mm/yr) as recorded by the G1 fans, exceed the estimated rock uplift rates of 0.4 – 0.6 mm/yr (Hilley and Strecker, 2005), and tectonic uplift alone is unlikely to be pulsed in a manner that would generate multiple fans. More likely, both climate forcing and tectonic forcing combine to produce and preserve the G1 fan sequence. Over the same period, curiously, no terraces are detected in the lower Toro Basin. Three



possible explanations for this absence (which are not mutually exclusive) include: (1) due to their more
central position within the basin, the lower reaches of the Río Toro were not strongly affected by rock
uplift, meaning that any changes in river-channel elevation are not perseveped in the geomorphic record
due to low or a lack of net incision; (2) channels in the lower Toro Basin continued to experience
aggradation or remained stable at this time, due to feedbacks in the system whereby incision upstream
caused a pulse of sediment for downstream reaches; or (3) the response time of the Río Toro within the
lower basin was substantially longer than the forcing period of the aggradation-incision cycles, meaning
perturbations to the channel-bed elevation due to climate forcing would not have reached so far
downstream.

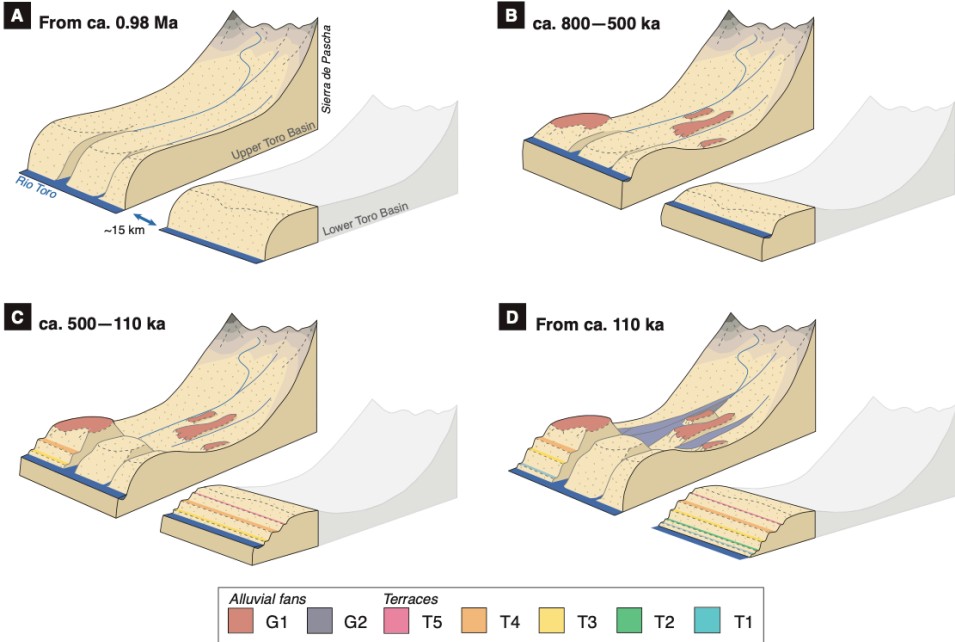


**Fig. 8.** Cartoon illustrating periods of aggradation and incision in the upper and lower Toro Basin from ca. 0.98 Ma. Area of
Lower Toro Basin block shaded in grey was not part of this study. A) From 0.98Ma: Base level lowered to present day levels,
following the deposition of Alfarcito Conglomerates. Renewed hydrological connectivity likely led to extensive sediment
evacuation and incision of (paleo)topography. Deposition of Quaternary Terrace Conglomerates started from 0.94 Ma (Hilley
and Strecker, 2005). B) ca. 800-500 ka: G1 fan formation and abandonment during a phase of net incision in the upper basin,
linked to the MPT. No significant geomorphic change recognised in the lower basin. C) ca. 500-110 ka: 100-kyr cycles of
aggradation and incision recorded by lower basin cut-and-fill terraces (T5, T4, T3). No significant geomorphic change
recognised in the tributaries of the upper basin. D) From ca. 110 ka: G2 fan formation and abandonment in the upper basin,
linked to ca. 21/40 kyr climate cycles. Continuation of 100-kyr cycles recorded by lower basin terraces (T2, T1).

To elaborate on the first possibility, the Sierra de Pascha catchments are positioned behind and
perpendicular to the axis of an elevated northward-plunging anticline (Fig. 2A). In concert with the
work by Hilley and Strecker (2005), we suggest that channel incision through the resistant anticline
accelerated sometime between 0.98 and 0.8 Ma. Once this incision propagated upstream and to the east

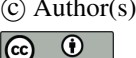



of the anticline, the removal of weakly consolidated sedimentary units in the upper basin was likely
efficient (Hilley and Strecker, 2005). The evolving topography of the anticline could therefore help to
explain the net incision needed in the upper Toro Basin to preserve the alluvial fan surfaces between ca.
800 to 500 ka, and why terrace levels in the lower basin are not recognised during this time interval.

To elaborate on the second and third possibilities, as late Quaternary glaciations were limited to the
Pascha tributary headwaters (< 5 km from headwall), the hillslope geomorphic response to prolonged
and intensified glaciation may have been very localized (Tofelde et al., 2018). This is apparently true
for the Iglesia and Calingasta Basins in the Western Precordillera where the tributaries, rather than the
main basin record incision following the Mid-Pleistocene Transition (Terrizzano et al., 2017; Peri et al.,
2022). Following this argument, the response time of the Río Toro's long profile to the 100-kyr climate
cycles after the Mid-Pleistocene Transition (ca. 1.2 to 0.8 Ma) may have been substantially longer than
the period of external forcing. If true, this implies that while upstream reaches of the channel may have
experienced no (or a very low amplitude) aggradation/incision cycles (Allen, 2008; McNab et al., 2023).
Alternatively, feedbacks within the system could lead to differences not only in the magnitude of
aggradation/incision, but also the timing. For example, in southwest Peru, Steffan et al. (2009, 2010),
interpreted aggradation in downstream reaches of river channels during past wet climate periods to
result from pulses of sediment mobilized from hillslopes and upstream channel incision.

*5.2.2 Terrace formation from ca. 500 to 110 ka*
From ca. 500 to 110 ka in the upper Toro Basin, we find no record of fan formation (Fig. 8). Curiously
again, though, the lower Toro Basin exhibits a spectacular sequence of terraces showing 100-kyr
cyclicity starting from ca. 500 ka (Tofelde et al., 2017). If long channel response times explain the lack
of terraces from ca. 800 – 500 ka in the lower Toro Basin, to explain the terraces identified in the lower
basin ca. 500 ka (Tofelde et al., 2017), the channel response time must have changed. This could have
occurred as a result of incision in the upper Toro Basin, which would have narrowed the upstream river
valleys, consequently decreasing river response times and enabling aggradation-incision cycles to affect
channel reaches farther downstream (e.g., McNab et al., 2023).

While a shortened channel response time can explain the formation of terraces in the lower Toro Basin,
it does not explain the absence of terraces/fans in the upper basin over the same period. Consequently,
we next consider other factors that might lead to differences in fan/terrace preservation between the
upper and lower Toro basins.

Perturbations at the Río Toro outlet, such as a shift in base level, will propagate upstream over time,
thus driving the net incision needed to preserve variations in channel bed elevation in the terrace and
fan sequences. Alternatively, activity along the Gólgota Fault at this time may have adjusted the base



level for the trunk stream. Regardless of the exact trigger for base-level fall (e.g., renewed fluvial
connectivity, possibly enhanced by a drop in Lerma Valley lake level) (Malamud et al., 1996; González
Bonorino and Abascal, 2012), a net incisional wave would have propagated upstream from the lower
basin or outlet. That incision would have facilitated terrace preservation in the lower Toro Basin before
the incisional wave propagated upstream to the upper Toro Basin. Steepened reaches of both the trunk
stream and tributaries up to an elevation of ca. 3400 m (Fig. 2C) are consistent with an upstream
propagating wave of incision, which probably only recently reached the ca. 3300-m elevation of the G2
fan toes.
Consistent with this interpretation, both the upper and lower Toro basins preserve geomorphic evidence
of channel-bed elevation lowering after ca. 100 ka (terraces T2 and T1 in the lower Toro Basin; G2 fan
generation in the upper Toro Basin). Whereas T2 and T1 lie 40 m and 20 m respectively above the
modern Río Toro, the G2 fans are at most 10 m above their closest channel. This finding further supports
the idea that net incision is ongoing in the lower Toro Basin, probably keeping pace with the ongoing
uplift of the Sierra de Pascha Sur (Tofelde et al., 2017), but net incision has possibly only resumed
within the last ca. 110 to 50 kyr in the upper Toro Basin.
Other factors may have also played a role in the misaligned timing of fan/terrace formation in the upper
and lower Toro basins. Restricted hydrological connectivity or disconnectivity can lead to internal
variability in the nature and timing of a basin's geomorphic or sedimentary response to external
perturbations (Fryirs et al., 2007; Buter et al., 2022). For example, basin connectivity and geometry
appear to have disrupted the timing of climate-driven sediment transfer within the Humahuaca Basin of
NW Argentina during the last glacial cycles, leading to anti-phased timing of aggradation-incision
cycles along tributaries on either side of the valley (Schildgen et al., 2016). No fault lines, which can
influence connectivity (Guarnieri and Pirrotta, 2008; Brocard et al., 2012), intersect the channel network
between the alluvial fans and terrace levels of the Toro Basin (Fig 2) (Pingel et al., 2020). Nevertheless,
minor adjustments to the long profile of an alluvial channel network can be sufficient to affect the
internal connectivity of a basin (Savi et al., 2020). One such adjustment may include the tributary
junction fan at the Quebrada de Chorro outlet, which has created a diffuse knickzone in the Río Toro
long profile (Fig. 2B). As the fan has aggraded, it has pushed the main channel to the opposite valley
side, evidenced by a marked channel bend. The fan may therefore inhibit the coupling between the
upstream and downstream reaches of the trunk stream by disrupting the flow of sediment and (possibly)
water from the Sierra de Pascha tributaries and along the Río Toro (e.g., Harvey 2012). However, the
capacity of the fan to disrupt environmental signals moving through the basin may depend on the
direction of signal travel. For example, channel incision due to a climate-induced increase in water
discharge may continue to propagate downstream, regardless of a new sedimentary input from a major
tributary, unless the tributary fully dams the upstream section. However, if a wave of incision is instead



migrating upstream, a tributary junction fan may slow or disrupt its propagation (Savi et al., 2020).
Nevertheless, while sedimentary inputs from individual tributaries can affect the modern channel
profile, and may slow upstream-propagating incisional cycles, it is not clear whether such localized
features will play an important role in channel network evolution over longer (e.g., > 100 kyr)
timescales.

*5.2.3 Fan formation since ca. 110 ka*
All G2 surfaces were either stable or actively receiving sediment for some time during both cool, wet
glacial periods and warm, dry interglacials. Similar to the terraces in the lower basin (Tofelde et al.,
2017), the timing of G2 surface abandonment is restricted to glacial phases; enhanced moisture
availability due to an intensified SASM is likely to have amplified sediment transport and channel
incision (Baker and Fritz, 2015). Around the latitude of the Toro Basin, glacial moraine records in the
Central Andes show strong evidence for glacial advances at ca. 16 and 22–24 ka, with some evidence
also for advances at ca. 42 and 55 ka (D'Arcy et al., 2019a; Fig 7B). The stratigraphically highest
surfaces in G2, Qf_5 and Qf_6, show abandonment ages that are consistent with the timing of the oldest
glacial advances recorded in the moraine record (ca. 55 ka).

For surfaces Qf_7 and Qf_8, the timing of abandonment is harder to interpret, due to the difficulty in
knowing whether the youngest boulders on each surface are outliers due to erosion/rotation, or if they
represent a time of active deposition on the surface. Given the similarities in surface weathering between
Qf_6 and Qf_7, it is possible that Qf_7 was active at the same time as Qf_6 and Qf_5, and hence was
abandoned at a similar time (implying that the youngest boulder of Qf_7 is an outlier). If the young
boulder instead represents a real depositional age, then the abandonment of Qf_7 could be linked to the
ca. 22–24 ka glacial advance, coinciding with the northern hemisphere Last Glacial Maximum. The
abandonment of Qf_8 is similarly challenging to interpret, with abandonment potentially linked to either
the ca. 24 ka glacial advance (associated with the 'Minchin' wet climate phase of the Central Andes) if
the youngest boulder is excluded, or the ca. 16 ka glacial advance associated with Heinrich Stadial 1 if
not excluded.

While we reason that the two youngest ages from Qf_7 and Qf_8 are not outliers and instead reflect
later deposition events (see 5.1.2), we have also estimated the timing of surface abandonment without
them (Fig. 7). In this alternative record, the abandonment of three of the four fans fall between ca. 65
and 60 ka. This points to a modest phase of net incision in several Sierra de Pascha catchments during
a wet glacial period (Fritz et al., 2007).





Overall, the exposure age distributions and estimated abandonment ages appear to capture cycles of fan
aggradation-incision with a periodicity of ca. 20 to 40 kyr. Considering the above tentative links
between abandonment times and glacial advances, and that no known tectonic forcing in the Toro Basin
can explain this cyclicity, the alluvial channel network is likely responding to precession (21-kyr) or
obliquity-driven (40-kyr) climate cycles. Precessional forcing has been recorded within the sedimentary
archives elsewhere in the Central Andes, including fluvial terraces in the Humahuaca Basin (23°S)
(Schildgen et al., 2016) and alluvial fans in the Santa María Basin (26.5°S) (D'Arcy et al., 2018) in NW
Argentina.

**5.3 Impacts of the Mid-Pleistocene Transition on the Toro Basin**

The G1 fan surfaces have CRN exposure ages that span several glacial-interglacial cycles (Fig. 6).
Although our interpreted ages are too imprecise to associate with specific glacial phases, 100-kyr glacial
moderation of aggradation-incision cycles is thought to have controlled fluvial terrace formation in the
lower Toro Basin (e.g., Tofelde et al., 2017). In semi-arid landscapes and transport-limited systems, this
finding is not unexpected, as geomorphic activity is invariably amplified during wetter, glacial periods
(Harvey et al., 1999; Spelz et al., 2008; Cesta and Ward, 2016). Given the number of G1 fans (n=4)
capturing the prolonged net incisional phase (>300 kyr), it is possible that eccentricity-driven cycles of
aggradation and incision are also recorded in the upper Toro Basin.

Our net incisional phase between ca. 800 and 500 ka coincides with the onset of prolonged and enhanced
global glacial cycles following the Mid-Pleistocene Transition (MPT, 1.2–0.8 Ma) which marked a shift
in climate periodicity from 41 to 100 kyr cycles (Berger et al., 1999). The southward migration of the
ITCZ at this time led to heightened moisture availability throughout the Central Andes (Haselton et al.,
2002; Broccoli et al., 2006; Vizy and Cook, 2007). Alluvial channels in semi-arid regions of the Central
Andes are found to respond quickly to marked shifts in precipitation such as this (e.g., Schildgen et al.,
2016; Tofelde et al., 2017), which also appear to drive phases of enhanced sediment evacuation to the
foreland (Fisher et al., 2023).

Enhanced incision linked to the MPT has also been recognised at other locations in the Central Andes
(Fig. 1A), including the Casa Grande Basin (23°S) in the Eastern Cordillera, the Salinas Grandes Basin
(23.5°S) of the Puna Plateau (Pingel et al., 2019b), and the Iglesia (30.5°S) and Calingasta (32°S) basins
in the Western Precordillera (Terrizzano et al., 2017; Peri et al., 2022). These observations point to a
regional phase of net incision and therefore landscape response to global climate change. For several of
these locations, including the Toro Basin, local tectonic activity may have provided a secondary driver
for incision, or created conditions conducive to fan/terrace preservation. Towards the Andean interior,
the geomorphic response to the MPT probably lessens, as moisture and the extent of past glaciations is



more restricted (Luna et al., 2018; Haselton et al., 2002). Beyond the Central Andes, fluvial terraces
along the Río Deseado (47°S) (Tobal et al., 2021) and Río Santa Cruz (50°S) (Milanez Fernandes,
2023), draining the Southern Andes in Patagonia also record a period of net incision that can be
tentatively linked to the MPT. On a global scale, a growing number of studies have identified periods
of intensified erosion at this time, for example in the St. Elias mountains, Alaska (Gulick et al., 2015),
Central Appalachia (Del Vecchio et al., 2022), the Rocky Mountains (Pederson and Egholm, 2013) and
the European Alps (Haeuselmann et al., 2007; Valla et al., 2011; Sternai et al., 2013). While it is not
possible to discount a tectonic influence on landscape change in the upper Toro basin entirely, the links
between MPT climate and incision, and its expression elsewhere in the Andes and beyond, is
compelling.

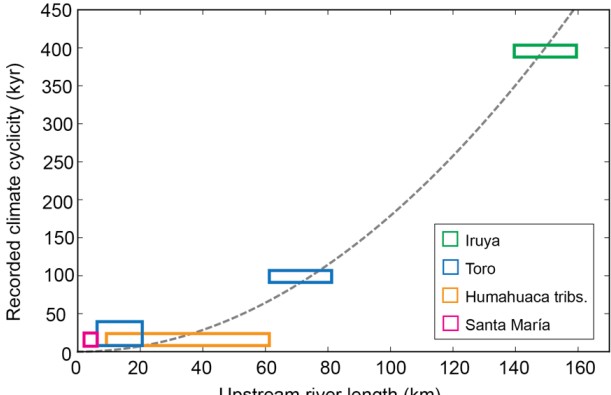


**Figure 9.** Correlation between recorded climate cyclicity and upstream river length recognised in four basins of the Central
Andes (Fisher et al., 2016; Schildgen et al., 2016; Tofelde et al., 2017; D'Arcy et al. 2018; this study). Adapted from Tofelde
et al. (2017). Recorded period: $0.019*\text{river length}^2$.

## 5.4 Climate periodicity and alluvial channel system length

Higher frequency climate cycles are recorded in fan generation G2 of the Sierra de Pascha tributaries
compared with the mainstem of the basin; the alluvial fans, which appear to record climate cycles with
a periodicity of ca. 20 to 40-kyr have an upstream channel length of ~10 km and are positioned ~30 km
upstream of the terrace sequence showing 100-kyr climate cyclicity dated by Tofelde et al. (2017). This
finding substantiates the theory that the response time of alluvial channel systems to perturbations in
climate depends on system length (Paola et al., 1992; Castelltort and Van Den Driessche, 2003; Godard
et al., 2013; McNab et al., 2023). Evidence of this relationship, together with the dependency on the
square of the system length, was identified in the archive of several sedimentary basins in the Central
Andes, although only a single forcing frequency was recorded within each basin (Fig. 9) (Tofelde et al.,
2017). Our new data from the Toro Basin provide critical field evidence that multiple climate



periodicities can be preserved within the sedimentary record of a single sedimentary basin, with higher forcing frequencies recorded only in the uppermost reaches of the basin.

## 6. Conclusions

The alluvial fan and terrace sequences of the Toro basin present an excellent opportunity to explore (1) how channel responses to external perturbations may or may not propagate downstream, and (2) the differences in landscape response to forcing frequency as a function of stream length. We applied CRN dating to a suite of alluvial fan surfaces to characterise the evolution of the alluvial channel network of the Toro basin over the last one million years. Our key findings are as follows:

1. We identified two generations of fan surfaces (G1 and G2) were identified in the Sierra de Pascha tributary catchments. The G1 fans record CRN exposure ($^{10}$Be) ages between ca. 800 and 500 ka, whereas the G2 fans record surface activity and then abandonment between ca. 100 and 20 ka.

2. The G1 fans capture a significant phase of net incision (~ 200 m) between ca. 800 and 500 ka. The stepwise evacuation of the upper basin coincides with the onset of prolonged and enhanced global glacial cycles following the Mid-Pleistocene Transition (MPT). With several basins in the Central Andes and beyond also registering this phase of incision, we propose that the G1 fans are part of a continental scale response to MPT climate change.

3. The abandonment of the G2 fans is restricted to glacial periods, mostly modulated by 21/40-kyr climate cycles; enhanced moisture availability due to an intensified SASM likely amplified channel incision and sediment transport.

4. Differences in the timing of alluvial fan and fluvial terrace development in the upper and lower Toro basins appear to be associated with how channel length affects fluvial response time to climate forcing as well as local controls on net incision, which facilitates preservation of the geomorphic record of aggradation-incision cycles.

5. The new alluvial fan record from the upper Toro Basin, combined with earlier results on fluvial terraces from the lower Toro Basin, provides field evidence for the theoretical predictions of a scaling relationship between climate forcing frequency recorded in sedimentary archives and the system length. We show that multiple climate periodicities can be preserved within the sedimentary record of a single sedimentary basin, with higher forcing frequencies recorded only in the uppermost reaches of the basin. This improved understanding of the role of system length in climate signal propagation is an important step forward in helping us to anticipate the spatial distribution of sedimentary paleoclimate records within landscapes.



### 7. Code/data availability

All data is included as part of the manuscript.

### 8. Author contribution

Conceptualization: E.N.O, T.F.S, S.T; Sample collection and processing: E.N.O, T.F.S, S.T, H.W.; Visualization: E.N.O with feedback from all authors; Writing & editing: all authors.

### 9. Competing interests

The authors declare that they have no conflict of interest.

### 10. Acknowledgments

This work was co-funded by (1) the German Research Foundation (DFG) grant 373/34-1 and the Brandenburg Ministry of Sciences, Research, and Cultural Affairs, within the framework of the International Research Training Group IGK2018 SuRfAce processes, TEctonics and Georesources: The Andean foreland basin of Argentina (StRATEGy) and (2) the European Research Council (ERC) under the European Union's Horizon 2020 Research and Innovation program (ERC Consolidator Grant 863490 to T.F.S.). TanDEM-X 12-m resolution digital elevation data were provided by the German Aerospace Center (DLR) through grant DEM_GEOL1915 to T.F. S. We thank Yanina Rojo for logistical support leading up to and during all field work. We also thank Peter van der Beek for assistance during field work.

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
