# Peer review of "in the southern Central Andes"

_EGUsphere, 2024_

## Referee Comment (RC3)

**Review of «Landscape response to tectonic deformation and cyclic climate change since ca. 800 ka in the southern Central Andes" - Orr. et al.**

**General comments**:

- **Scientific significance (Good)**: the paper discusses the importance of alluvial records in capturing climate variability relative to the length of rivers within an orogenic system, specifically in broken foreland setting. This topic is intriguing because the organization and spatial patterns of alluvial terraces and fans may be indicative of various driving factors, including tectonic activity, climatic changes, and autogenic processes. The results presented in the manuscript add valuable insights into the interpretation of alluvial sequences during the late Pleistocene.

- **Scientific quality (Good)**:
  - Methods:
    - The authors have mapped alluvial fans using satellite imagery along with DEMs and have dated these features using Cosmogenic Radionuclide 10Be derived from surface samples and a depth profile. One can regret that the analysis of the surface samples did not also incorporate 26Al, as this dual measurement could have provided a more comprehensive discussion of the laboratory errors or potential sample inheritance issues that might affect the 10Be age estimates.
    - The authors should introduce here how the depth profile provides more reliable results and how to use the results versus the surface exposure age.
    - Are there any erosional rates available for this area? If so, this could impact the CRN ages.
    - Furthermore, the authors distinguish between the ages of the alluvial fans' aggradation and their ages of abandonment. This aspect is quite interesting, but the method by which the abandonment ages were determined remains unclear to me. Although the authors cite earlier studies, it would be beneficial to elaborate on the methodology and clarify the uncertainties associated with these results, as they play a crucial role in the discussions presented in the paper.
    - Eventually, in Figures 6C and 7C, the authors display areas of probable age for the terraces using the Probability Density Functions. Could you remind here the concept?
  - Results: I find the section 4 dedicated to the results, not well adjusted. The section starts with a conclusion of the results comparing two generations of fans (G1 and

G2). This helps to split the description of the results in two subsections dedicated to G1 fans and then G2 fans but it is confusing in the reading as all information and discussion are given at the same time.  You have first to show that you are working on alluvial fans, then you date them. I would rewrite the section 4 as follow:

- Section 4.1: a description of the alluvial fans morphology, arrangement (cut and fill, fill-cut, strath), sediment types/size, slope illustrated with pictures (Qf_1 and Qf_5 are not shown on Figure 3) from the field or extracted from satellite images. In this section, it might be good to have a longitudinal incision profile of the alluvial fans versus their nearest stream (height of alluvial fans with regards to the stream bed rock versus distance to the river outlet with significant vertical exaggeration) since you give those values in the text. This could fit in Figure 3.
- Section 4.2: CRN ages of the surface activity and abandonment of the individual alluvial fans described in 4.1 and discuss potential inheritances/erosional factors.
- The identification of two generations of fans is either a conclusion to this section 4 (4.3) or part of the discussion.

o Discussion: The discussion presented in the paper is engaging but occasionally difficult to follow, largely due to the absence of visual aids to clarify the various hypotheses being examined. In my view, the paper's findings predominantly indicate a renewed phase of incision after 750 ka, which the authors attribute to the shift in the duration of glacial cycles during the Middle Pleistocene Transition (1.2-0.8 Ma). Additionally, the research points to a depositional hiatus between 500 and 100 ka in the tributaries of the lower Toro Basin, whereas during the same period, alluvial terraces were developing along the Toro River in the upper basin. To explain this spatio-temporal variabilities, the authors work on three hypotheses: (i) tectonic uplift in the upper reaches, (ii) upstream/downstream incision/aggradation feedbacks, (iii) response time to base level variabilities due to climate variabilities.

- The authors favor hypothesis (iii) but the impact of the base level variability in the Lerma Valley at the Toro Basin outlet has been poorly investigated. There is a possibility that autogenic processes or the reorganization of the drainage network may control a sudden drop in base level. Additionally, the contribution of tectonic uplift at the basin's outlet versus the river erosion power, as detailed by Hilley and Strecker (2005), needs consideration. If these two processes sustain a sufficiently elevated base level, the system's response time could be significantly prolonged, which would give predominance to hypothesis (ii).

- The authors dismiss hypothesis (i), yet this requires further clarification since tectonic uplift rates comprise 50 to 80% of the incision rates. Consequently, a question arises as to why tectonic uplift couldn't be responsible for multiple generations of alluvial fans. It would be beneficial for the authors to explain this reasoning in more detail.
- To aid in comprehending the various hypotheses, I recommend including additional figures:
  - A table that succinctly compares the periods of activity and abandonment of the alluvial fans with the ages of the downstream terraces as identified in Tofelde et al. (2017).
  - A composite figure that visually represents the three hypotheses could be particularly helpful. This could take the form of schematic longitudinal profiles extending from the lower Toro Basin to the Lerma Valley, including the downstream tectonic barriers. The figure should illustrate how the longitudinal profile has evolved over time and how these changes correspond to the alluvial records. We should not rule out that all three hypotheses may contribute simultaneously.

One aspect that requires further clarification is the rationale behind the G2 fans reflecting climate periodicity of 20 to 40 kyr, as outlined in section 5.4. While initially introduced as a tentative hypothesis, this conclusion seems to be presented later as a more definitive outcome. Additionally, this premise is based on the constrained age distribution of the G2 fans, which is specified as between 21 and 40 kyr in section 5.1.2. However, there is a discrepancy, as the ages from CRN dating and the derived abandonment ages indicate a narrower span, with differences between terraces typically ranging from 5 to 20 kyr.

To clarify, section 5.1.2 should delve deeper into addressing the uncertainties associated with the ages, which could include potential inheritance effects or erosion impacts. It should also expound upon the aggradation activity duration, the precise timing of terrace abandonment, and how these periods correlate with global climate benchmarks such as Marine Isotope Stages (MIS), which have been utilized for the G1 fans' chronology.

- **Presentation quality** (**Fair**): as stated above, I would recommend improving or restructuring some sections (results and discussion). I would restructure the section *3 Methodology* as follow:

  *3.1 Mapping of the alluvial fans (from the field, satellite imagery and DEM)*
  *3.2 CNR dating*

*3.2.1 10Be surface exposures*
*3.2.2 Depth Profile*
*3.3 Surface abandonment (detail more d'Arcy et al., 2019)*

Also, I would also clarify some figures
o   Figure 1: The primary structural elements (thrusts, axes of anticlines/synclines) should be depicted in Figure 1B. This addition would clarify where tectonic uplifts might be anticipated and indicate the locations of active tectonic barriers to rivers. Also, what are the meaning of TRMM2B3, SASM and MPT (pentagon symbols) in the caption? In the section on climate setting or within the figure caption, please specify which glacial records are being referenced. If these records are not addressed in the text, they should be omitted from the figure to avoid confusion.
o   Figure 2:
- The color coding on the geological map (Figure A) could be made clearer—perhaps reducing the transparency would help.
- It would be useful to include the slope values for the Rio Toro segments and its tributaries in Figure B. These are key geomorphic metrics and would aid in comparing this study with others.
- The relevance of the knickpoint highlighted in the close-up of Figure B needs clarification. Is the delayed incision due to variability in bedrock or river morphometrics? This point warrants discussion in the text. Additionally, the paper would benefit from a satellite image of the Toro basin, delineating all the alluvial fans and terraces, similar to what is shown in Figure 4 of Hilley and Strecker (2005).
o   Figure 3:
- I would recommend to add a close-up satellite image of the alluvial fans.
- Add geographical orientations on the sections.
- Specify in the caption the type of projection on the section (orthogonal, distance?).
o   Figure 5: The picture of the transect is too smal. It would be worth to make it bigger.
o   Figure 6:
- The CaCO3 graph in Figure 6A appears to be inconsistent with the age of the G1 fans; thus, it may be unnecessary to include this graph.
- There does not seem to be any mention of Figure 6B in the text. If this figure isn't utilized, it should be removed to maintain clarity.
- In Figure 6C, distinct symbols should be used to mark the samples suspected to be affected by erosion or inheritance, as well as those that are stratigraphically inconsistent. Implementing this would better emphasize the conclusion that the alluvial fans were actively forming between 750 and 500 ka.
- I question the relevance of referencing Marine Isotope Stages (MIS) in this context since the age uncertainties associated with the fans surpass the resolution of MIS periods. Moreover, it does not appear that MIS are mentioned elsewhere in the text.
- Could you specify the meaning of PDF in the caption?
o Figure 7: the ages of 60-65 ka determined for the surface activity and abandonment of the Qf_5-7 seem to correspond with Marine Isotope Stage 4 (MIS4). This connection is noteworthy and should be mentioned.
o Figure 8: Could you provide the ages of the individual alluvial fans generation and terraces in the legend?
o Figure 9: Could you clarify which references pertain to the various boxes and highlight which results are from your own study?

I have added additional comments to the pdf files.

I am sorry for all those detailed comments, but I think that they will help increasing the quality of the manuscript.

Grégoire Messager

[revised manuscript text omitted]

---

## Author Response (AR1)

**Response to Reviewers Document**

**REVIEWER 1**

**General Comments**

1. **This manuscript by Orr et al. builds on a rich history of quality work done by this group tackling similar problems across the NW Argentinean Andes with the aim of using the rich geomorphic and tectonic settings of the region to constrain landscape responses and timing to climatic and tectonic perturbations. The paper is well written, reasoned, cited, and beautifully illustrated across all the figures. I really appreciate the care that this group of authors has taken with this work and think it should be accepted. The manuscript builds on previous work and datasets in a rigorous way to provide evidence for the preservation of varying geomorphic periodicities and processes within the same basin. I think the authors should be proud of their work.**

Thank you very much for your through review and constructive feedback on our manuscript. We have carefully addressed all of your comments, clarifying a number of aspects of the project. We believe that these adjustments have helped us to improve the manuscript – thank you!

2. **The only weakness I find in the work is inherent to the methods and certainly no fault of the authors. Exposure dating of surfaces is just difficult, and even when you are as careful as these authors, there are just so many unknowns that can affect the CRN ages. This is especially apparent with the G1 fans. The authors rightfully acknowledge the messiness of these data and derive an incision rate from 800 ka to 500 ka that they argue is related to the MPT. The problem is that you have to explain away a lot of ages either through inheritance for the older ones or erosion for the younger ones. This in turn leaves you wondering which ones are reliable and how do you know? The authors state in line 516 that they "suggest that the G1 fan record can instead be used to capture an extended phase of net incision within the Sierra de Pascha tributaries." Given the uncertainty in the ages in general it is hard to say that this incision occurred over 300 kyr or over 20 kyr. The point being that they invoke the MPT but without real good constraints you could invoke any number of drivers of incision over that period of time.**

Thank you for your feedback. Working with an age dataset with such broad age distributions is a challenge. By resampling the Qf_1 depth profile and integrating it with the new boulder ages, we were able to constrain the age of the oldest G1 fan surface. This then served as a benchmark for the rest of the dataset, enabling us to interpret the remainder of the fan record with increased confidence. Based on the fan stratigraphy and our observations of the surfaces and boulders (see Supplement 2 and 3), we identified the boulder which were likely affected by erosion (TB19_06, TB19_08, TB19_14) or inheritance (TB19_02, TB19_04). This enabled us to constrain a phase of net incision (ca. 800- 500 ka), during which each of the G1 fans were active at some point, before being abandoned. As stated in the manuscript, applying [21]Ne to some of these surfaces may help to reveal the complexities in burial history.

We acknowledge that there remain some uncertainties about the ca. 800-500 ka net incisional phase. Importantly, we are not suggesting a period of continuous incision during this time. Given our confidence in the Qf_1 dating (and that the other surfaces broadly fit into the a. 800-

500 ka window), we argue that the G1 fans nicely capture a phase of net incision, before a period of lower geomorphic activity. The onset of G1 fan activity and abandonment appears to coincide with the MPT. We feel as though this argument is compelling because similar phases of net incision linked to the MPT are recorded throughout the Andes. Throughout the manuscript, we acknowledge that there are other drivers during this time that may have contributed to the net incision. However, we have revised the wording in this section to clarify our arguments.

L515: 'Given these complexities in the fan chronostratigraphy, rather than identifying discrete phases of aggradation and incision for each fan surface, we suggest that the G1 fan record can instead be used to capture a phase of net incision within the Sierra de Pascha tributaries. Crucially, this is unlikely to be a phase of continuous incision, but rather a phase of net incision lasting ca. 300 kyr or less'.

3. **My only recommendation would be to temper the wording a bit (things like "likely") with respect to large-scale climate drivers as there is a fair amount of ambiguity that results from this technique and the correlations between fan age climate proxies in Figure 7 are not always obvious. I would tend toward phrases like "we propose" as the authors have done in their key finding #2 in the conclusions and acknowledge that there is considerable ambiguity.**

Thank you for the recommendation. As advised, we have adjusted the wording slightly to acknowledge remaining uncertainties in the results. We feel that Section 5.3 approaches this argument cautiously and draws on evidence from other studies to suggest a regional event. Rather than saying that the MPT has driven the incision entirely, we argue that there is compelling evidence of a potential link between them. We also acknowledge that other drivers (e.g. tectonic activity) may also contribute here. We have adjusted a couple of paragraphs to emphasise that we are being cautious with our interpretations.

L724: 'Enhanced incision likely linked to the MPT has also been recognised at other locations in the Central Andes (Fig. 1A), including the Casa Grande Basin (23°S) in the Eastern Cordillera, the Salinas Grandes Basin (23.5°S) of the Puna Plateau (Pingel et al., 2019b), and the Iglesia (30.5°S) and Calingasta (32°S) basins in the Western Precordillera (Terrizzano et al., 2017; Peri et al., 2022).'

L738: 'While it is not possible to discount a tectonic influence on landscape change in the upper Toro basin entirely due to some ambiguity in the datasets, the links between MPT climate and incision, and its expression elsewhere in the Andes and beyond, is compelling.'

**Specific Comments:**

4. **Any hypothesis why there is such a discrepancy between boulder dimensions between G1 and G2? Do the authors think this is systematic (source area difference?) or just a sampling bias? Is it evidence for widespread erosion of the G1 boulders?**

The G1 fan surfaces appeared more weathered and eroded (with some boulder spallation) than G2. This is expected, given that the surfaces are over ca. 400 kyr older and likely explains why the G1 boulders were noticeably smaller than those of G2. The boulder size is unlikely to reflect a difference in source area. We made a concerted effort to sample large, tabular, stable boulders that stood tall of the fan surface, with minimal evidence of weathering and surface spallation. While erosion may have affected a minority of the boulders (see response to Q2),

our careful sampling strategy means that we can be more confident that the boulder ages reflect periods of surface activity and/or stability.

5. **Figure 6 - I guess part A is informative for the reader but any comparison with the fan record in C is sort of laughable. Maybe consider what it adds to the story.**

Thank you for the feedback. We believe that it is important to include this data in the figure as it reinforces our argument that assigning G1 surfaces individually to particular forcing or events is not appropriate (see L457). Fig. 6 also includes some important geomorphic and isotope data which is used to explore some of the alternative drivers of incision. Overall we think this figure is appropriate, and helps to capture some of the climate, tectonic and geomorphic history of the basin on these very long timescales.

6. **Is it possible to put glacial extent in the Pascha headwaters on Figure 1B since you discuss local glaciations on line 592? Not sure but it looks like Martini et al., 2017 and Luna et al., 2018 could be put in there and give some spatial context for this part of the discussion.**

The past glacier extents in the tributaries have not been mapped or dated, although this presents an exciting avenue for future research. While the studies by Martini et al. (2017) and Luna et al. (2018) offer excellent glacier chronologies for local basins, we are very hesitant to extrapolate their data and apply their observations to the Toro Basin. Instead, we suggest that in line with regional glacial records (D'Arcy et al., 2019a), the Pascha tributaries were likely glaciated for periods in the last ca. 100 kyr.

7. **Figure 8B - You say "no significant geomorphic change recognized in the lower basin" but it appears in the diagram that there has been considerable mainstem Rio Toro aggradation from panel A to B. I think it is important to also detail what is going on along the mainstem in this figure as this is creating the local baselevel and presumably propagating any climatic signal between the two.**

Thank you for catching this error! The terraces record significant aggradation in the lower basin at this time (as reflected in the figure and later discussion). The local base level rose in line with this aggradation. The phrase "No significant geomorphic change in the lower basin" refers to the absence of an incision record. This has been clarified in the figure caption.

L573: 'Aggradation was recorded in the lower basin (Tofelde et al., 2017).'

8. **Line 693 - Doesn't this fall in a drier period based on the panel A in Figure 7? It is a period of lower insolation which seems to correlate with warmer sea surface temp and less negative delta 18O. Have a look and see if you agree.**

Thank you for catching this omission. We have adjusted this statement to reflect your comments.

L693: 'This points to a modest phase of net incision in several Sierra de Pascha catchments during a dry interglacial period (Fritz et al., 2007).'

9. **Conclusion point 3 - I don't understand the data that is backing up the statement "the abandonment of the G2 fans is restricted to glacial periods". In Figure 7 QF_5 and QF_6 both have pdfs that mostly predate any glacial ages. Is the idea that the onset of glaciation drove abandonment of these?**

Thank you for this comment. D'Arcy et al. (2019a) argue that glaciation in this region is broadly in phase with insolation cycles, periods of SASM strengthening and/or northern hemispheric events. While there are no glacial records local to Toro that recognise a glacial event between 70 and 60 ka, the climatic conditions were sufficient to sustain glaciers. Glaciers were also recorded throughout the Central Andes at this time (D'Arcy et al., 2019a).

**10. Also what constrains the very long tail of the glacial PDF if there are no points out there? Is that curve taken from a more extensive regional study? You should state in the caption what constrains this PDF.**

For clarity, we have removed the long tail on the glacier PDF.

**11. Conclusion point 5 - I think it needs to be pointed out in section 5.4 that the Rio Iruya dataset is different from the others in Figure 9 because it is not measuring the timing of formation or abandonment of a geomorphic feature but rather a nearly continuous chemical sediment signal of erosion from a catchment. I am still wrapping my head around what it all means in terms of comparing them but I think it is an important distinction to make.**

Thanks for highlighting this and we agree that this is an important distinction to make. We have added this clarification in the Fig. 9 caption.

L744: 'Unlike the other records of aggradation and incision, the Iruya record is derived from the basin's sedimentary record and is a paleo-erosion dataset.'

**Technical Corrections:**

**12. Line 520 - You say ~0.07 mm/yr of incision between 800 ka and 500 ka but then at line 557 it is 0.8 mm/yr. Just need to decide on the best value and be consistent.**

Thank you for catching this error. This incision rate is now quoted throughout the manuscript as 0.7 mm/yr.

**13. Figure 9 - Fisher et al, 2016 should be Fisher et al., 2023 in the caption or the citation needs to be added for the 2016 AGU talk. The 2023 article is better to cite than the 2016 AGU talk though.**

The Fisher et al. (2023) publication is now cited throughout the manuscript.

**14. Fisher et al, 2023 reference is missing the last author in the reference. Should be…
, and Lourens, L.J.**

The reference has now been revised to include the last author.
* * *
**REVIEWER 2**

1. **Congrats on this interesting study. Here are some minor comments meant to improve this work in the paragraphs related with the geological setting.**

Thank you very much for reviewing our manuscript and providing some helpful feedback on the sections relating to Toro's geological setting. We have addressed each of your comments and made the necessary changes to the revised manuscript.

2. **Line 123: I wonder what DeCelles et al., 2011 doing here, as they have not studied nor mentioned the course and nature of the Rio Toro.**

Thanks for catching this error. We have removed the citation.

3. **Lines 129-132: possibly site primary literature, or indicate by „reviewed in"**

Thank you for the suggestion. We have added the Alonso (1992) citation, which helped to inform the adapted geological map in Fig. 2A.

Alonso, R.N. Estratigrafía del Cenozoico de la cuenca de Pastos Grandes (Puna Salteña) con énfasis en la Formación. Revista de la Asociación Geológica Argentina, 47(2), 189-199.1992.

4. **Lines 132-134 „The Solá fault has been active since at least the Pliocene, and tectonic deformation from the Miocene to mid-Pleistocene has been recorded along the San Bernardo and Gólgota faults (Marrett and Strecker 2000)." Comment: ca. 10-9 Ma growth strata in the Agujas Cgl. in the Qda Cerro Bayo, along the Sola Fault indicate fault activity since at least Late Miocene times. Similarly, pre-13 Ma growth strata within the Barres Sst along the Golgota Fault set the minimum timing for deformation in the area (see DeCelles et al., 2011; Pearson et al., 2013; Pingel et al., 2020).**

Thank you for highlighting these omissions. We have adjusted the paragraph as recommended. We have also added the additional citations.

L135: 'The Solá fault has been active since at least the Late Miocene, and tectonic deformation from the Miocene to mid-Pleistocene has been recorded along the San Bernardo and Gólgota faults (Marrett and Strecker 2000; DeCelles et al., 2011; Pearson et al., 2013; Pingel et al., 2020). The Gólgota fault reactivated after ca. 0.98 Ma (Hilley and Strecker 2005)'.

5. **Lines 157-159: (1) There is no major N-trending anticline separating the Río Toro from the fan deposits, as indicated here in the text and in the geological map in Fig. 2. The topographic expression causing this separation is formed by west-tilted basin strata (Barres; Agujas; volcanics of the Las Burras-Almagro-El Toro magmatic complex; and Alfarcito Conglomerates), likely deformed via basin-internal shortening in response to contraction along the Golgota fault. The map in Fig. 2 itself shows no repetition of the strata, only westward younging, as would be expected for W-tilted strata.**

Thank you very much for your comment. We have carefully reviewed the geologic maps of the basin and agree with your argument. We have therefore removed the mapped anticline from Fig. 2A and adjusted the manuscript text.

L163: 'The Middle Miocene Barres Sandstone and Agujas Conglomerates, interbedded with lava flows, and the Pliocene-Pleistocene Alfarcito Conglomerates make up the west-tilted strata, which lay between the fan deposits and the Río Toro (Fig. 2A; Hilley and Strecker, 2005; Mazzuoli et al., 2008).'

L616: 'To elaborate on the first possibility, the Sierra de Pascha catchments are positioned behind and perpendicular to west-tilted and deformed basin strata (Barres Sandstone, Agujas and Alfracito Conglomerates, lava flows) (Fig. 2A). In concert with the work by Hilley and

Strecker (2005), we suggest that channel incision through the resistant sedimentary units accelerated sometime between 0.98 and 0.8 Ma. Once this incision propagated upstream, the removal of weakly consolidated sedimentary units in the upper basin was likely efficient (Hilley and Strecker, 2005). This evolving topography could therefore help to explain the net incision needed in the upper Toro Basin to preserve the alluvial fan surfaces between ca. 800 to 500 ka, and why terrace levels in the lower basin are not recognised during this time interval.'

6. **Something is wrong with the Mio-Pliocene basin stratigraphy: The Agujas Conglomerates are missing entirely from the record. The middle Miocene Barres sst is overlain by the Late Miocene Agujas Cgl, followed by the Alfarcitos cgl. ; all units are intercalated with volcanics of various kind.**

Thank you for catching this oversight. We have adjusted the legend of Fig 2A and included the Agujas Conglomerate in the main text.

L163: 'The Middle Miocene Barres Sandstone and Agujas Conglomerates, interbedded with lava flows, and the Pliocene-Pleistocene Alfarcito Conglomerates make up the west-tilted strata, which lay between the fan deposits and the Río Toro (Fig. 2A; Hilley and Strecker, 2005; Mazzuoli et al., 2008).'Resistant Barres, Agujas and Alfarcito units characterise several erosional surfaces that stand ~700 m above the modern river channel'.

7. **The literature presented to underpin the commented statement is partly inadequate: a) DeCelles et al.'s part on the Toro Basin is from the Qda. Cerro Bayo along the Sola Fault near Est. Maury not discussing the upper Toro Basin; b) Robledo et al. is a paleontological work dealing with flora and insect trace fossils of a Mio-Pliocene section along the Sola Fault near Gobernador Solá; c) Mazzuoli et al., is the only reference that fits, as they have dated the volcanics in the center of the ridge. None of the sources presented mention an anticline.**

Thank you for your comments. We have removed the relevant citations.

8. **Line 171: „South American low-level jet (SALLJ)" in Fig. 2 it is called LLAJ–Low-Level Andean Jet.**

This has been changed in the revised manuscript.

9. **Label for Almagro Range is positioned wrong.**

The position of the label has been adjusted.

10. **The legend states that the unit below the Miocene volcanics is the Barres Fm. This is not correct. It is mainly the Agujas Conglomerates plus the underlying Barres sandstone.**

Thank you for catching this. We have adjusted the legend to be consistent with the geological map in Pingel et al., (2020) (Fig. 3).
* * *
**REVIEWER 3**

**General Comments**

1. **Methods:**

a. **The authors have mapped alluvial fans using satellite imagery along with DEMs and have dated these features using Cosmogenic Radionuclide 10Be derived from surface samples and a depth profile. One can regret that the analysis of the surface samples did not also incorporate 26Al, as this dual measurement could have provided a more comprehensive discussion of the laboratory errors or potential sample inheritance issues that might affect the 10Be age estimates.**

We agree that there are many opportunities for future research and that pairing our new $^{10}$Be record with other radionuclides (e.g. $^{21}$Ne [as outlined in the manuscript], $^{26}$Al) in the future would help to further scrutinise Toro's sedimentary archive. Unfortunately, this is beyond the scope of the present study. However, preliminary work to do this is in progress (https://doi.org/10.5194/egusphere-egu24-14778). Those preliminary results, moreover, do not change any of the fundamental conclusions that we reach in this manuscript, as we have strived to be conservative in our interpretations here.

b. **The authors should introduce here how the depth profile provides more reliable results and how to use the results versus the surface exposure age.**

Thank you for highlighting this point. Our aim was to pair the resampled $^{10}$Be depth profile results with the boulder exposure ages to robustly constrain the timing of fan surface activity. We do not wish to claim that one approach is more robust than the other, but rather that their combination are likely to better capture the nuances of the surfaces' exposure histories. Given the timescales considered, and the geological uncertainties involved, we avoid stating that the depth profile results are more reliable. Below are excerpts from the manuscript detailing how the depth profile results are used to build the Toro fan chronostratigraphy.

L354: 'The Qf_1 $^{10}$Be depth profile, using combined $^{10}$Be data from this study and from Tofelde et al., (2017), was used to determine an exposure age using the Hidy et al. (2010) Monte Carlo simulator. Further details are provided in Supplement 1 and 2'.

L501: 'Pairing the Qf_1 $^{10}$Be depth profile with the surface boulder exposure ages means that we can more robustly constrain the oldest phase of fan development within the study area and use it as a benchmark when evaluating the remainder of the G1 fan record'.

c. **Are there any erosional rates available for this area? If so, this could impact the CRN ages.**

This is an important question. This very old semi-arid landscape records very low erosion rates within several of the tributary catchments (<0.03 and 0.12 mm/yr) (Tofelde et al., 2018). We sampled the largest and freshest boulders on each surface. For this reason, and consistent with nearby work by Schildgen et al. (2016) and Tofelde et al. (2017), we assume zero erosion when calculating the $^{10}$Be boulder ages. For the $^{10}$Be depth profile, a range of erosion rates were considered in the Hidy simulations (see Supplement 1). This is to account for fan surface inflation and depression.

L249: 'Today, catchment-averaged erosion rates for catchments draining the Sierra de Pascha range between <0.03 and 0.12 mm/yr (Tofelde et al., 2018).'

S_L45: 'The basin's climate and our careful sampling strategy meant that corrections for snow cover and boulder erosion were not necessary (Schildgen et al., 2005, 2016; Tofelde et al., 2017).'

S_L62: 'The erosion rate (cm/ka) was set to range between -0.2–0.2 and the total erosion threshold (cm) was set to -10–50. These negative values for erosion simulate inflation (Hidy et al., 2010)'.

We acknowledge the potential complexities of the surfaces' exposure histories, which may include boulder surface erosion. However, without sufficient data to constrain this, we do not think it is appropriate to further model the timing of landform abandonment. This has motivated new work which is applying $^{21}$Ne to the boulder samples to better decipher the exposure histories of the fan surfaces and provide some constraints on boulder surface erosion rates.

L334: 'Working with chronological data at this coarse resolution over $10^{5}$–$10^{6}$-year timescales means that even the most sophisticated inheritance/erosion models are limited in their ability to estimate the timing of landform abandonment (e.g., Prush and Oskin, 2020; Dortch et al., 2022).'

L501: 'Pairing the $^{10}$Be record with cosmogenic $^{21}$Ne in the future may help to decipher some of the complexities in the exposure histories of the boulders; $^{21}$Ne is well suited for quantifying long term landscape change in arid, low erosion environments (Dunai et al., 2005; Ma and Stuart, 2018)'.

d. **Furthermore, the authors distinguish between the ages of the alluvial fans' aggradation and their ages of abandonment. This aspect is quite interesting, but the method by which the abandonment ages were determined remains unclear to me. Although the authors cite earlier studies, it would be beneficial to elaborate on the methodology and clarify the uncertainties associated with these results, as they play a crucial role in the discussions presented in the paper.**

Thank you for highlighting this—we agree that more information could be provided here. We have clarified the approach in the main text.

L325: 'The probabilistic model for inferring the timing of fan surface abandonment from D'Arcy et al. (2019 b) was applied to fans with exposure ages of less than ca. 300 ka. The model uses the exposure ages of boulders on the fan surface to generate a probability distribution of abandonment ages and a most probable abandonment age. The modelled abandonment age is based on the premise that an alluvial fan surface remains active for a period of time that may generate a range of exposure ages exceeding the uncertainty bounds on any individual age. The calculated abandonment age and its uncertainty is thus dependent on the youngest measured exposure age, the duration of surface activity, and the number of samples. For a detailed description of the approach, see D'Arcy et al. (2019b).'

e. **Eventually, in Figures 6C and 7C, the authors display areas of probable age for the terraces using the Probability Density Functions. Could you remind here the concept?**

PDFs are used to help visualise the age distributions of the surfaces and is consistent with similar literature. Given the large age distributions, we do not use the PDF modelled ages to outline the timing of fan activity (i.e. a specific age), but we do think it is important to include the PDFs in Figs. 6 and 7. The PDFs for the G2 fan abandonment do include specific abandonment ages (shown by vertical lines within inverted PDFs in Fig 7C.)

2. **Results: I find the section 4 dedicated to the results, not well adjusted. The section starts with a conclusion of the results comparing two generations of fans (G1 and G2). This helps to split the description of the results in two subsections dedicated to G1 fans and then G2 fans but it is confusing in the reading as all information and discussion are given at the same time. You have first to show that you are working on alluvial fans, then you date them. I would rewrite the section 4 as follow:**
   i. **Section 4.1: a description of the alluvial fans morphology, arrangement (cut and fill, fill-cut, strath), sediment types/size, slope illustrated with pictures (Qf_1 and Qf_5 are**

not shown on Figure 3) from the field or extracted from satellite images. In this section, it might be good to have a longitudinal incision profile of the alluvial fans versus their nearest stream (height of alluvial fans with regards to the stream bed rock versus distance to the river outlet with significant vertical exaggeration) since you give those values in the text. This could fit in Figure 3.

    ii. **Section 4.2: CRN ages of the surface activity and abandonment of the individual alluvial fans described in 4.1 and discuss potential inheritances/erosional factors.**

    iii. **The identification of two generations of fans is either a conclusion to this section 4 (4.3) or part of the discussion.**

Thank you for this feedback. We put a lot of thought into how to best structure the results as clearly and concisely as we could. We believe that it is important to summarise the results first, to clearly justify why we have divided the Toro fan surfaces into two generations. This approach – to address each fan generation in turn – is the way the remainder of the results and then discussion sections are structured. Establishing this structure early on allows us to provide detailed descriptions of the landforms/ages later, while also highlighting methodological, analytical, and interpretative differences between the generations. For this reason, we have opted to retain the current structure.

Figure 2C shows the tributary long profiles alongside the projected fan surfaces. In the results section, we specify the elevation of the surfaces above the closest modern channel. Channel locations are further confirmed in Figure 3B. Instead of adding another figure/panel to an already lengthy paper, we believe that we have already provided the information requested.

3. **Discussion:**
   a. **The discussion presented in the paper is engaging but occasionally difficult to follow, largely due to the absence of visual aids to clarify the various hypotheses being examined. In my view, the paper's findings predominantly indicate a renewed phase of incision after 750 ka, which the authors attribute to the shift in the duration of glacial cycles during the Middle Pleistocene Transition (1.2-0.8 Ma). Additionally, the research points to a depositional hiatus between 500 and 100 ka in the tributaries of the lower Toro Basin, whereas during the same period, alluvial terraces were developing along the Toro River in the upper basin. To explain this spatio-temporal variabilities, the authors work on three hypotheses: (i) tectonic uplift in the upper reaches, (ii) upstream/downstream incision/aggradation feedbacks, (iii) response time to base level variabilities due to climate variabilities.**

       i. **The authors favor hypothesis (iii) but the impact of the base level variability in the Lerma Valley at the Toro Basin outlet has been poorly investigated. There is a possibility that autogenic processes or the reorganization of the drainage network may control a sudden drop in base level. Additionally, the contribution of tectonic uplift at the basin's outlet versus the river erosion power, as detailed by Hilley and Strecker (2005), needs consideration. If these two processes sustain a sufficiently elevated base level, the system's response time could be significantly prolonged, which would give predominance to hypothesis (ii).**

Thank you for your thoughtful feedback. As you highlight, we have provided a number of possible explanations for the observed events and patterns in the data, and we walk through the rationale for why we favour some interpretations over others. Therefore, we outline as clearly as possible, based on the evidence available, the potential roles of autogenic forcing, drainage reorganization, and tectonic uplift in shaping the basin's evolution. One point of apparent confusion is that three hypotheses are considered specifically with respect to explaining the lack of terrace generations in the lower Toro Basin over the time period which the G1 fan levels are created in the upper Toro Basin (see 5.2). This may be the reason why the reviewer believed we discounted a role for base level fall at the valley outlet, when, to the contrary, we believe base level fall (triggered by some process) is critical for explaining the preservation of the downstream terrace sequence. We have attempted to make the text as concise and well-structured as possible, walking readers through these complex arguments.

We have also added a new table (Table 3, L625), which summarises the evolution of the lower and upper basin.

With regards to autogenic forcing, we already noted the following:

L581: "While autogenic processes, such as channel avulsion and meander cut-offs may also play a role in channel incision and the formation of discrete fan lobes or terraces (Nicholas and Quine, 2007; Ventra and Nichols, 2014), we consider the scale of channel incision associated with the features of interest (ranging from ca. 10 to hundreds of meters) is beyond the scope of purely autogenic behavior."

Also, like Tofelde et al. (2017) we do indeed consider that base-level fall, which may have occurred as a result of a number of different processes, to be the driver for net incision that ultimately helps preserve the different terrace levels, at least within the lower Toro Basin (see L621-639).

Finally, an elevated base level has no effect on the system response time, as this is primarily a function of system length, water discharge, sediment discharge, and valley width (see for example McNab et al., 2023, and earlier references noted there). Base-level change is obviously a potential trigger for response (i.e., channel incision), which we discuss in the lines noted above (L691).

> **b. The authors dismiss hypothesis (i), yet this requires further clarification since tectonic uplift rates comprise 50 to 80% of the incision rates. Consequently, a question arises as to why tectonic uplift couldn't be responsible for multiple generations of alluvial fans. It would be beneficial for the authors to explain this reasoning in more detail.**

Thank for the raising these concerns. On the contrary, we argue that tectonic uplift may have played a role in the net incisional events recorded, and that the interplay between climate and tectonics is likely responsible for the record preserved.

L246: 'Tofelde et al. (2017) thus concluded that while the renewed uplift of the Sierra de Pascha Sur helped initiate the deposition of the Terrace Conglomerates, the periodicity of the cut-and-fill cycles is best explained by orbitally driven climate forcing, with net incision likely associated with the channel response to ongoing rock-uplift'.

L592: 'However, another mechanism is likely at play because the averages rates of incision between ca. 800 and 500 ka (0.8 mm/yr) as recorded by the G1 fans, exceed the estimated rock uplift rates of 0.4 – 0.6 mm/yr (Hilley and Strecker, 2005), and tectonic uplift alone is unlikely to be pulsed in a manner that would generate multiple fans. More likely, both climate forcing and tectonic forcing combine to produce and preserve the G1 fan sequence'.

L832: 'While it is not possible to discount a tectonic influence on landscape change in the upper Toro basin entirely due to some chronological ambiguity in the datasets and inherent challenges in deconvolving different forcing mechanisms, the links between MPT climate and incision, and its expression elsewhere in the Andes and beyond, is compelling'.

For the G1 fans, we argue that the cyclical nature of the record cannot be explained by tectonics alone.

L789: 'Considering the above tentative links between abandonment times and glacial advances, and that no known tectonic forcing in the Toro Basin can explain this cyclicity, the alluvial channel network is likely responding to precession (21-kyr) or obliquity-driven (40-kyr) climate cycles.'

**c. To aid in comprehending the various hypotheses, I recommend including additional figures:**

  **i. A table that succinctly compares the periods of activity and abandonment of the alluvial fans with the ages of the downstream terraces as identified in Tofelde et al. (2017).**

The PDFs in Figs 6 and 7 are included for easier comparison of the timing of fan surface activity and abandonment. We are cautious about specifying time periods for G1 fans due to the large age distributions and associated uncertainties (L482). For this reason, we believe that it is not appropriate to include them. The caption for Fig. 7 does provide the modelled abandonment ages. Fig. 8 is, nevertheless, a schematic representation of when activity was occurring in the upper and lower Toro Basin. However, we agree that understanding would be aided with a summary table, which we have included in the revised manuscript (Table 3, L625). Thank you for the suggestion.

  **ii. A composite figure that visually represents the three hypotheses could be particularly helpful. This could take the form of schematic longitudinal profiles extending from the lower Toro Basin to the Lerma Valley, including the downstream tectonic barriers. The figure should illustrate how the longitudinal profile has evolved over time and how these changes correspond to the alluvial records. We should not rule out that all three hypotheses may contribute simultaneously.**

Thank you for the suggestion – we agree that creating figures to explain key concepts or arguments is a good thing to do. Fig. 8 shows how the basin has evolved over the last 1Ma in specific stretches of the upper and lower reaches of the basin. We did consider developing a composite figure as suggested, but it is very difficult to clearly and accurately create a visualisation like this that reflects all hypotheses outlined. For example, while we are able to plot how the profile of specific reaches of the basin changed at specific timesteps, we are not able to do this consistently throughout the study area. Instead, we have included long profiles and the projected fan surfaces in Figs 2 and 3, which can be used in conjunction with Fig. 8 and Table 3 to think about how the elevations of the Toro drainage network has evolved over time. Moreover, the three hypotheses only reflect one part of the basin evolution – specifically, the lack of terrace generation downstream while fans were generated upstream.

**4. One aspect that requires further clarification is the rationale behind the G2 fans reflecting climate periodicity of 20 to 40 kyr, as outlined in section 5.4. While initially introduced as a tentative hypothesis, this conclusion seems to be presented later as a more definitive outcome. Additionally, this premise is based on the constrained age distribution of the G2 fans, which is specified as between 21 and 40 kyr in section 5.1.2. However, there is a discrepancy, as the ages from CRN dating and the derived abandonment ages indicate a narrower span, with differences between terraces typically ranging from 5 to 20 kyr.**

Thank you for this constructive feedback. We have toned down some of the language in the latter sections of the manuscript to better reflect the certainty of our arguments. Some of the G2 fan surfaces may have been active and then abandoned within the same climate cycle. Also, we agree that while the abandonment ages of the fan surfaces reflect relatively high periodicity forcing, we can only speculate that that may be related to precessional or obliquity cycles. We have modified the wording of the text to better reflect that uncertainty.

L788: 'Overall, the exposure age distributions and estimated abandonment ages appear to capture cycles of fan aggradation-incision with relatively high periodicity. Considering the above tentative links between abandonment times and glacial advances, and that no known tectonic forcing in the Toro Basin can explain this potential cyclicity, the alluvial channel network is possibly responding to precession (21-kyr) or obliquity-driven (40-kyr) climate cycles'.

L877: 'The abandonment of the G2 fans is restricted to glacial periods, possibly modulated by 21/40-kyr climate cycles; enhanced moisture availability due to an intensified SASM may have amplified channel incision and sediment transport.

5. **To clarify, section 5.1.2 should delve deeper into addressing the uncertainties associated with the ages, which could include potential inheritance effects or erosion impacts. It should also expound upon the aggradation activity duration, the precise timing of terrace abandonment, and how these periods correlate with global climate benchmarks such as Marine Isotope Stages (MIS), which have been utilized for the G1 fans' chronology.**

In the methods and results sections, we detail some of the possible sources of uncertainty in the $^{10}$Be record, and on L526-559, we note why we believe inheritance plays a smaller role in the G2 fan generation compared to the G1 generation. But without additional data, it is difficult to delve deeper into these uncertainties, and we prefer to avoid pure speculation. Given the current data resolution, we are also cautious about attempting to correlate cycles of aggradation and incision with other records (e.g., MIS) beyond what we have already done. We believe such correlations would not enhance the value of our interpretations.

6. **Fig. 1: The primary structural elements (thrusts, axes of anticlines/synclines) should be depicted in Figure 1B. This addition would clarify where tectonic uplifts might be anticipated and indicate the locations of active tectonic barriers to rivers. Also, what are the meaning of TRMM2B3, SASM and MPT (pentagon symbols) in the caption? In the section on climate setting or within the figure caption, please specify which glacial records are being referenced. If these records are not addressed in the text, they should be omitted from the figure to avoid confusion.**

Thank you for this feedback on Fig. 1. The primary structural elements are depicted in detail in Fig. 2A. We did attempt to add some of the structures to Fig. 1B, but unfortunately the figure became too busy and difficult to decipher. Instead, we have opted to label the mountain ranges only, which are then described in section 2.1. TRMM2B3, SASM and MPT are now defined in full in the figure caption. The glacial records referenced in Fig. 1 are referenced in the caption. These records are referred to throughout the remainder of the manuscript (e.g., discussion around Fig. 7). Reference to Fig. 1A is now made in the Fig. 7 caption.

7. **Fig. 2: The color coding on the geological map (Figure A) could be made clearer—perhaps reducing the transparency would help. - It would be useful to include the slope values for the Rio Toro segments and its tributaries in Figure B. These are key geomorphic metrics and would aid in comparing this study with others. - The relevance of the knickpoint highlighted in the close-up of Figure B needs clarification. Is the delayed incision due to variability in bedrock or river morphometrics? This point warrants discussion in the text. Additionally, the paper would benefit from a satellite image of the Toro basin, delineating all the alluvial fans and terraces, similar to what is shown in Figure 4 of Hilley and Strecker (2005).**

Thank you for these suggestions. The transparency of the geological unit polygons is to ensure that the underlying topography remains visible. Given the importance of the topography in our interpretations, we felt that this was important to emphasise. The knickzone highlighted in Fig. 2B is explained in section 5.2.2 (L743). A tributary junction fan at this location may have inhibited the coupling between the upstream and downstream reaches of the Toro.

Satellite imagery can be a useful way to present basin geomorphology. At the scale which includes both the fans and terraces, it is very difficult to distinguish between landforms. For this reason, we opted to include the maps, profiles and cross sections in Figs. 1 and 2. Crucial

elevation data is presented in these figures (that the satellite imagery could not capture). For this reason, we decided that satellite imagery would not significantly add information to the manuscript and has therefore been omitted. We think the photographs of the fan surfaces in Fig. 4 also help the reader to visualise the geomorphic setting of the upper basin.

8. **Fig. 3. I would recommend to add a close-up satellite image of the alluvial fans. - Add geographical orientations on the sections. - Specify in the caption the type of projection on the section (orthogonal, distance?).**

Thank you for these suggestions. While we acknowledge that satellite imagery can be used effectively to present a landscape, we have opted to prioritise the maps, profiles and cross sections that we have included. Please see our response to Comment 8. The geographical orientations of the sections are shown by projection lines and labels in Fig. 3A. The projection type is now included in the caption.

9. **Fig. 5. The picture of the transect is too small. It would be worth to make it bigger.**

We assume that the 'transect' refers to the field image of the depth profile. This image is scaled to match the depth profile axis on the left. For this reason, the figure remains unchanged in the revised manuscript.

10. **The CaCO3 graph in Figure 6A appears to be inconsistent with the age of the G1 fans; thus, it may be unnecessary to include this graph. - There does not seem to be any mention of Figure 6B in the text. If this figure isn't utilized, it should be removed to maintain clarity.**

Thank you for these suggestions. We have included the Lake Titicaca record because it is one of the few environmental records in the area that extends beyond 100 ka, and it was used by Tofelde et al. (2017) to help decipher the drivers of the cut-and-fill terrace cycles shown in Fig. 6B. Given that part of the G1 record overlaps with these records, we felt that it was important to include.

**In Figure 6C, distinct symbols should be used to mark the samples suspected to be affected by erosion or inheritance, as well as those that are stratigraphically inconsistent.**

We have tried to keep the figures as simple as possible. In the main text we detail the samples that we suspect have nuances to their exposure history and then reference Fig. 6. We are hesitant to change the symbology of the samples, because there remains some uncertainty about the nature of these nuances (ongoing work aims to better understand some of this better).

**I question the relevance of referencing Marine Isotope Stages (MIS) in this context since the age uncertainties associated with the fans surpass the resolution of MIS periods. Moreover, it does not appear that MIS are mentioned elsewhere in the text. - Could you specify the meaning of PDF in the caption?**

The isotope record in Fig. 6 serves as a global climate record and allows us to reference the Mid-Pleistocene Transition (MPT) in an accessible way. A PDF is a probability density function and is defined in the figure caption.

11. **Figure 7: the ages of 60-65 ka determined for the surface activity and abandonment of the Qf_5-7 seem to correspond with Marine Isotope Stage 4 (MIS4). This connection is noteworthy and should be mentioned.**

We have modified this text to respond to a reviewer's comment.

L785: 'In this alternative record, the abandonment of three of the four fans fall between ca. 65 and 60 ka. This points to a modest phase of net incision in several Sierra de Pascha catchments during a dry interglacial period (Fritz et al., 2007).'

**12. Figure 8: Could you provide the ages of the individual alluvial fans generation and terraces in the legend?**

Thank you for this suggestion. We have added these ages to the caption.

**13. Figure 9: Could you clarify which references pertain to the various boxes and highlight which results are from your own study?**

Thank you for this suggestion. We have adjusted the caption as requested.

**Specific Comments (SC)**

**SC1-L84:** Comment addressed in C7.

**SC2-L85:** Comment addressed in C7.

**SC3-L89:** Comment addressed in C7.

**SC4-L96:** Vertical resolution of DEM is now quoted in caption of Fig. 1.

**SC5-L213:** External drainage is drainage beyond the basin's limits and into the foreland.

**SC6-L219:** The drivers of this external base-level change are not certain. Further discussion is available in Seagren and Schoebohm, (2021). Further details of this change are therefore not included within the revised manuscript.

**SC7-L224:** Thank you for this suggestion. As the terrace record is not the main focus of this study, we opted instead to summarise their location and stratigraphy in Fig. 2 and 8 only. However, maps and block diagrams capturing the flight of terraces are available in Tofelde et al. (2017).

**SC8-L230:** Thank you for this question. We have adjusted the sentence to improve clarity.

L246: 'Moreover, the calculated net incision rate through the terrace sequence of 0.4 mm/yr from ca. 500 ka is consistent with long term rock-uplift rates of the Sierra de Pascha Sur (Hilley and Strecker, 2005).'

**SC9-L241:** Acronym has been defined on L117.

**SC10-L257:** Vertical resolution of DEM is defined in Fig. 1 (L99). SC4- L96.

**SC11-L258:** Please see response to comments 8 and 9.

**SC12-L262:** DEM is now defined in L279.

**SC13-L263**: GRASS and GDAL are toolboxes available in QGIS.

**SC14-L264:** QGIS is a geographic information system software. This has been clarified in the revised manuscript.

**SC15-L264:** Acronym has been defined on L117.

**SC16-L303:** Acronym is now clearly defined in revised manuscript.

**SC17-L320:** For the Hidy simulations we set the erosion rate range between -0.2–0.2 mm/yr.

S_L62: 'The erosion rate (cm/ka) was set to range between -0.2–0.2 and the total erosion threshold (cm) was set to -10–50. These negative values for erosion simulate inflation (Hidy et al., 2010)'.

**SC18-L332:** Thank you for this comment. This has been addressed in our response to comment 2b.

**SC19-L342:** Thank you for this comment. This has been addressed in our response to comment 3.

**SC20-L343:** Thank you for this comment. This has been addressed in our response to comment 3.

**SC21-L345:** As we outline above, the structuring of the Results took some thought. The G2 surfaces had to be mentioned here, as we had to draw a distinction between the surfaces early on. In the next paragraph (L378), the G2 fans are introduced in full.

**SC22-L355:** Thank you for this comment. This has been addressed in our response to comment 3.

**SC23-L392:** This information is already provided in Table 2 (L434).

**SC24-26-L394:** Further details about the Hidy model simulations are provided Supplement 1, Hidy et al. (2010) and Tofelde et al. (2017).

S_L62: 'The erosion rate (cm/ka) was set to range between -0.2–0.2 and the total erosion threshold (cm) was set to -10–50. These negative values for erosion simulate inflation (Hidy et al., 2010)'.

**SC27-L402:** Thank you for this suggestion. We felt that it was important to not include any data interpretation or discussion in the Results. For this reason, we wait until later sections to outline our explanations for Qf_1's age distribution.

**SC28-L411:** We do not define this unit as a strath surface in the manuscript.

**SC29-L414:** We define the sampled surfaces as 'alluvial fan surfaces'. Descriptions of the surfaces and sediment is provided from L364.

**SC30-L422:** The [10]Be datasets (Table 1, 2) are referenced in the early paragraphs of the Results.

**SC31-L424:** The 'most probable abandonment age' refers to the modelled age using the approach detailed in D'Arcy et al., (2019b). Further information about the approach is provided in the revised manuscript.

L329: 'The probabilistic model for inferring the timing of fan surface abandonment from D'Arcy et al. (2019 b) was applied to fans with exposure ages of less than ca. 300 ka. The model uses the exposure ages of boulders on the fan surface to generate a probability distribution of abandonment ages and a most probable abandonment age. The modelled abandonment age is based on the premise that an alluvial fan surface remains active for a period of time that may generate a range of exposure ages exceeding the uncertainty bounds on any individual age. The calculated abandonment age and its uncertainty is thus dependent on the youngest measured exposure age, the duration of surface activity, and the number of samples. For a detailed description of the approach, see D'Arcy et al., (2019b).'

**SC32-L427:** Information in the supplements doesn't really support this section of the main text.

**SC33-L441:** The PDF of this surface shows a bimodal age distribution (Fig. 7C). For this reason, we modelled abandonment ages with and without the youngest age.

**SC34-L447:** Please see response to comment 6.

**SC35-L491:** We opt to use lower case letters when referring to the probability density functions in the caption.

**SC36-L517:** This is an estimate of how much vertical incision has occurred between ca 0.98 Ma and ca. 800 ka based on the elevations of the paleotopography and the Qf_1 fan surface (see Fig. 3B, C).

**SC37-L520:** Thank you very much for this suggestion. We agree that there are lots of approaches to presenting our incision estimates. In an effort to keep the manuscript as concise as possible, we have opted to summarise this information in Fig. 3, 8 and Table 3. Since these incision rates are estimates and the fan surface ages have large distributions, we decided not to plot them against each other. We do not think that it would contribute anything new to the manuscript.

**SC38-L525:** Thank you for this feedback. These comments have been addressed as part of the 'General Comments' section.

**SC39-L559:** Cycles of fan activity and abandonment are unlikely the result of tectonic forcing alone.

L793: 'Overall, the exposure age distributions and estimated abandonment ages appear to capture cycles of fan aggradation-incision with relatively high periodicity. Considering the above tentative links between abandonment times and glacial advances, and that no known tectonic forcing in the Toro Basin can explain this cyclicity, the alluvial channel network is likely responding to precession (21-kyr) or obliquity-driven (40-kyr) climate cycles'.

**SC40-L584:** We have removed reference to an anticline in the revised version of this manuscript.

**SC41-L621:** There are a couple of studies that have discussed Pleistocene base level change.

L721: 'Regardless of the exact trigger for base-level fall (e.g., renewed fluvial connectivity, possibly enhanced by a drop in Lerma Valley lake level) (Malamud et al., 1996; González Bonorino and Abascal, 2012), a net incisional wave would have propagated upstream from the lower basin or outlet.'

**SC42-L623:** The geological map (including fault lines) has been adapted from Segemar 250k geological maps and Pingel et al. (2020). We have made the faults as clear as possible on the map, without detracting from other important details.

**SC43-L675:** Please see our response to comment 6 and 11.

**SC44-L675:** Thank you for these queries. Please see our response to comment 5.

**SC45-L675:** Please see our response to comment 5.

**SC46-L710:** 'Geomorphic activity' might include both phases of aggradation and incision, depending upon where you are in the basin.

**SC47-L712:** 300 kyr is the difference between ca. 800 ka and 500 ka (phase of net incision).

**SC48-L722:** Thank you for these citation recommendations. These are now included in the revised version of this manuscript.

**SC49-L751:** Please see our response to comment 5.